# Meta-analysis towards FSHD reveals misregulation of neuromuscular junction, nuclear envelope, and spliceosome
Teresa Schätzl[1], Vanessa Todorow[2], Lars Kaiser[1], Helga Weinschrott[1], Benedikt Schoser[2], Hans-Peter Deigner[1,3,4], Peter Meinke[2] & Matthias Kohl [1]✉

Facioscapulohumeral muscular dystrophy (FSHD) is one of the most common autosomal dominant muscle disorders, yet no cure or amelioration exists. The clinical presentation is diverse, making it difficult to identify the actual driving pathomechanism among many downstream events. To unravel this complexity, we performed a meta-analysis of 13 original omics datasets (in total 171 FSHD and 129 control samples). Our approach confirmed previous findings about the disease pathology and specified them further. We confirmed increased expression of former proposed *DUX4* biomarkers, and furthermore impairment of the respiratory chain. Notably, the meta-analysis provides insights about so far not reported pathways, including misregulation of neuromuscular junction protein encoding genes, downregulation of the spliceosome, and extensive alterations of nuclear envelope protein expression. Finally, we developed a publicly available shiny app to provide a platform for researchers who want to search our analysis for genes of interest in the future.

Facioscapulohumeral muscular dystrophy (FSHD) is an autosomal dominant inherited muscle disorder characterized by weakness and atrophy. It is one of the most common muscular dystrophies. According to the latest European epidemiological study (published in 2014), the prevalence of FSHD is 5–12 affected individuals per 100,000 population[1]. The pathomechanism of FSHD has not been fully elucidated yet, and no drug cures the disease or slows its progression.

A milestone in FSHD research was the discovery of the abnormal activity of a gene called *DUX4*, which is thought to be involved in regulating the cleavage phase of embryonic development[2], but is otherwise silenced throughout life, except for low-level expression in the thymus and testes[3]. In healthy individuals, it has been reported to be hypermethylated at its locus on chromosome 4q35, a macrosatellite consisting of 11–100 or more D4Z4 repeats, each containing a *DUX4* gene. In FSHD, this chromosomal segment is shortened to 10 or fewer repeats, with concomitant detection of hypomethylation allowing reading of the most distant *DUX4* gene. In combination with the 4qA haplotype, which contains a polyadenylation signal, *DUX4* can be expressed due to this contraction[4]. The encoded double homeobox protein 4 (DUX4) is a transcription factor that ultimately triggers signaling cascades by activating other transcription factors and hundreds of genes, resulting in rapid cell death[5]. Generally, there are two types of FSHD, which show exactly the same phenotype. Ninety-five percent of

patients with FSHD have FSHD1, and the remaining percentage have FSHD2. In FSHD2, there is no contraction of the D4Z4 repeats, as the 4q35 locus contains about 11–20 repeats. Instead, there is a mutation in either *SMCHD1* (>80% of FSHD2), *DNMT3B* or *LRIF1* gene, which leads to hypomethylation of the D4Z4 repeats and allows expression of *DUX4*[6].

The exact mechanisms of the signaling pathways prevalent in FSHD are still unclear. Further decelerating research progression and insight, FSHD is characterized by extreme variability in phenotype, even compared with other muscular dystrophies. This variability can be observed between affected family members and in a frequent body asymmetry, with individual muscle groups having different degrees of damage[7]. Moreover, although several studies have reported an inverse correlation between the number of D4Z4 repeats and disease severity[8,9], other studies examining few[10] or comparatively many repeats[11] have detected that D4Z4 allele size is not always related to clinical severity. In addition, the size of repeats in the upper range was reported to have lower methylation than predicted for a comparatively milder phenotype[12]. It suggests that unknown additional factors are involved, which may ultimately mitigate or exacerbate disease progression. This is further supported by the fact that pathological changes in muscle morphology and integrity generally start in the second decade of life[4] with varying muscle groups affected only slightly or not at all, hinting at rescue or compensation mechanisms preventing *DUX4* expression and toxicity.

[1]Institute of Precision Medicine, Furtwangen University, Furtwangen, Germany. [2]Friedrich-Baur-Institute at the Department of Neurology, LMU University Hospital, Ludwig Maximilian University, Munich, Germany. [3]Faculty of Science, Eberhard-Karls-University Tuebingen, Tuebingen, Germany. [4]EXIM Department, Fraunhofer Institute IZI (Leipzig), Rostock, Germany. ✉e-mail: matthias.kohl@stamats.de

Given the unknown factors of FSHD pathology, advances in transcriptomics technologies over the past decades harbor great potential in elucidating molecular mechanisms and pathogenic signaling pathways using bioinformatics approaches and statistical methods. In this context, to better understand the complexity of FSHD, we performed a meta-analysis (PROSPERO ID: CRD42022330489[13]) to verify existing knowledge and to identify additional characteristics that could advance FSHD research.

## Results

Our search identified a total of 11 studies, summarized in 13 datasets (Table 1), that met the predefined criteria based on data from databases, related publications, and information obtained during our literature search through email contact with the corresponding authors (Fig. 1a; detailed description of the individual phases of the meta-analysis can be found in Supplementary Notes 1, 2 and Supplementary Data 1, sheet 1; the PRISMA 2020 Checklist[14] and Checklist for Conducting Meta-Analysis of Microarray Datasets[15] are depicted in Supplementary Table 2). The stages of unified preprocessing and statistical analysis (Fig. 1b) culminated in the summary results of a random-effects model[16], which were confirmed in all subsequent areas by a secondary analysis approach, the vote-counting[15] (see the "Methods" section and Supplementary Notes 6). The random-effects meta-analysis yielded 1935 significant results (adjusted (adj.) $p$-value < 0.05). These results are decreasingly sorted by standardized mean difference with heteroscedastic population variances in the two groups (SMDH)[17] as shown in the heatmap in Fig. 1c, which is linked to data summarized in Supplementary Data 1, sheet 2a. The results are verified by sensitivity analyses (Supplementary Notes 3 and 4), enrichment analyses (random-effects model and vote-counting approach; Supplementary Data 1, sheets 3, 5 and 6), clustering of Gene Ontology (GO) terms using the Bioconductor package simplifyEnrichment (Supplementary Figs. 1–3), and the results of the meta-FSHD app we developed (see the "Methods" section).

### Meta-FSHD app

We have developed a publicly available shiny app called meta-FSHD to provide a tool for researchers to quickly and easily get the estimated meta-analytic effect for any gene of potential interest (Fig. 1d). The app considers all genes measured in at least three datasets, corresponding to 26,858 unique ENSEMBL-IDs, 21,080 unique ENTREZ-IDs, and 22,791 unique gene names. It gives detailed parameters regarding significance, effect size, confidence interval (CI), and degree of heterogeneity. Using meta-FSHD, all

results of the meta-analysis on significant genes can be confirmed and are traceable easily and quickly for any person (see the "Methods" section).

### Confirmation of previous knowledge of FSHD pathology

Several studies have already demonstrated that the clinical picture of FSHD is associated with the highly toxic expression of the *DUX4* transcription factor[18–20]. Since the gene is expressed both sporadically and in only a few myonuclei (1 in 200–1000 cells), detection is difficult[5], which is why DUX4 target genes are investigated[21]. In the meta-analysis, DUX4 biomarker genes are the most upregulated (Fig. 2a; Supplementary Data 1, sheet 2a). The highest expression is found in *H3Y1* (standardized log2-fold change (STD log2-FC) (95% CI) = +2.89); while histone variant *H3* was previously linked to *DUX4*[22], it was later stated that DUX4 induces *H3Y* and *H3X*, which mark DUX4 target genes for expression[23]. Besides, as shown by our GO clusters (Supplementary Fig. 1), there are enriched Biological Process (BP) sections of up- and downregulated genes related to development, differentiation, and morphogenesis. DUX4 generally has been described as a disruptor of muscle myogenesis that, when present at high levels, leads to apoptosis and, when present at lower levels, inhibits myogenesis[24]. In this context, the BP GO clusters and the gene list (Supplementary Data 1, sheet 2a) further show strong inflammasome activation and upregulation of apoptotic processes (Fig. 2b).

In addition, the meta-analysis shows that metabolic genes are misregulated in FSHD (Fig. 2c and BP clusters in Supplementary Fig. 1). Several studies have already pointed to mitochondrial abnormalities in FSHD[25,26]. A recent study was able to identify mitochondria as a source of excessive reactive oxygen species due to impairment of mitochondrial oxidative phosphorylation (OXPHOS), particularly in complex I, as an early event of DUX4-induced toxicity[27]. This is of great interest because in proliferating healthy myoblasts, approximately 30% of the ATP consumed by the cells is generated by OXPHOS. In contrast, in terminally differentiated myotubes, mitochondrial respiration is the major source of ATP (~60%)[28]. The impairment of the respiratory chain was described to lead to an immediate decrease in metabolic activity followed by a gradual increase in mitochondrial membrane potential. The general consequences observed were apoptosis by mitochondrial reactive oxygen species and impairment of mitochondrial health by lipid peroxidation[27]. Furthermore, it was reported that impaired metabolic adaptation would lead to misdirected increase in hypoxia signaling[27,29]. About these findings, we noticed a significant upregulation of *HIF1α* in 10 of the 13 datasets (STD log2-FC (95% CI) =

## Table 1 | Overview of datasets

|  | Author & Year | Dataset | Technology | FSHD[a] | CRTL |
|---|---|---|---|---|---|
| Microarrays | Cheli et al.[70] | GSE26061 | Affymetrix Human Exon 1.0 ST Array | 10 | 6 |
|  | Tsumagari et al.[71] | GSE26145 | Affymetrix Human Exon 1.0 ST Array | 6 | 6 |
|  | Osborne et al. (2008) | GSE10760 | Affymetrix Human Genome U133A & B Arrays | 19 | 30 |
|  | Arashiro et al.[73] | GSE15090 | Affymetrix Human Genome U133 Plus 2.0 Array | 5 | 5 |
|  | Rahimov et al.[74] | GSE36398 | Affymetrix Human Gene 1.0 ST Array | 13 | 12 |
|  | Tasca et al. (2012) | GSE26852 | Illumina HumanHT- 12 V3.0 expression beadchip | 8 | 7 |
| RNA-Seq | Yao et al. (2014) | GSE56787[b] GSE56787celllines | Illumina HiSeq 2500 | 15 | 9 |
|  |  |  |  | 9 | 5 |
|  | Wang et al. (2018) | GSE115650 | Illumina HiSeq 2500 | 33 | 9 |
|  | Banerji et al.[77] | GSE123468_12[c] GSE123468_16 | Illumina HiSeq 2500 | 6 | 6 |
|  |  |  |  | 6 | 6 |
|  | van den Heuvel et al.[69] | EGAD00001008337 | Illumina HiSeq 4000 & Illumina NovaSeq 6000 & NextSeq 500 | 37 | 24 |
|  | Watt et al.[79] | GSE138768 (DUX4)[d] | Illumina HiSeq 2500 | 4 | 4 |
|  |  |  | $\Sigma$ | 171 | 129 |

[a]More detailed information regarding patient and tissue characteristics, exclusion of samples and significant results can be found in Supplementary Table 1.
[b]GSE56787 included both cell lines and biopsies.
[c]GSE123468 included two families (two sisters each as patient and control), which resulted in family-related batch effects in our statistical analysis due to strong genetic similarity in the families.
[d]GSE138768 contains data from an artificial DUX4-model and was therefore excluded from the overall calculation, but was contrasted for comparison.

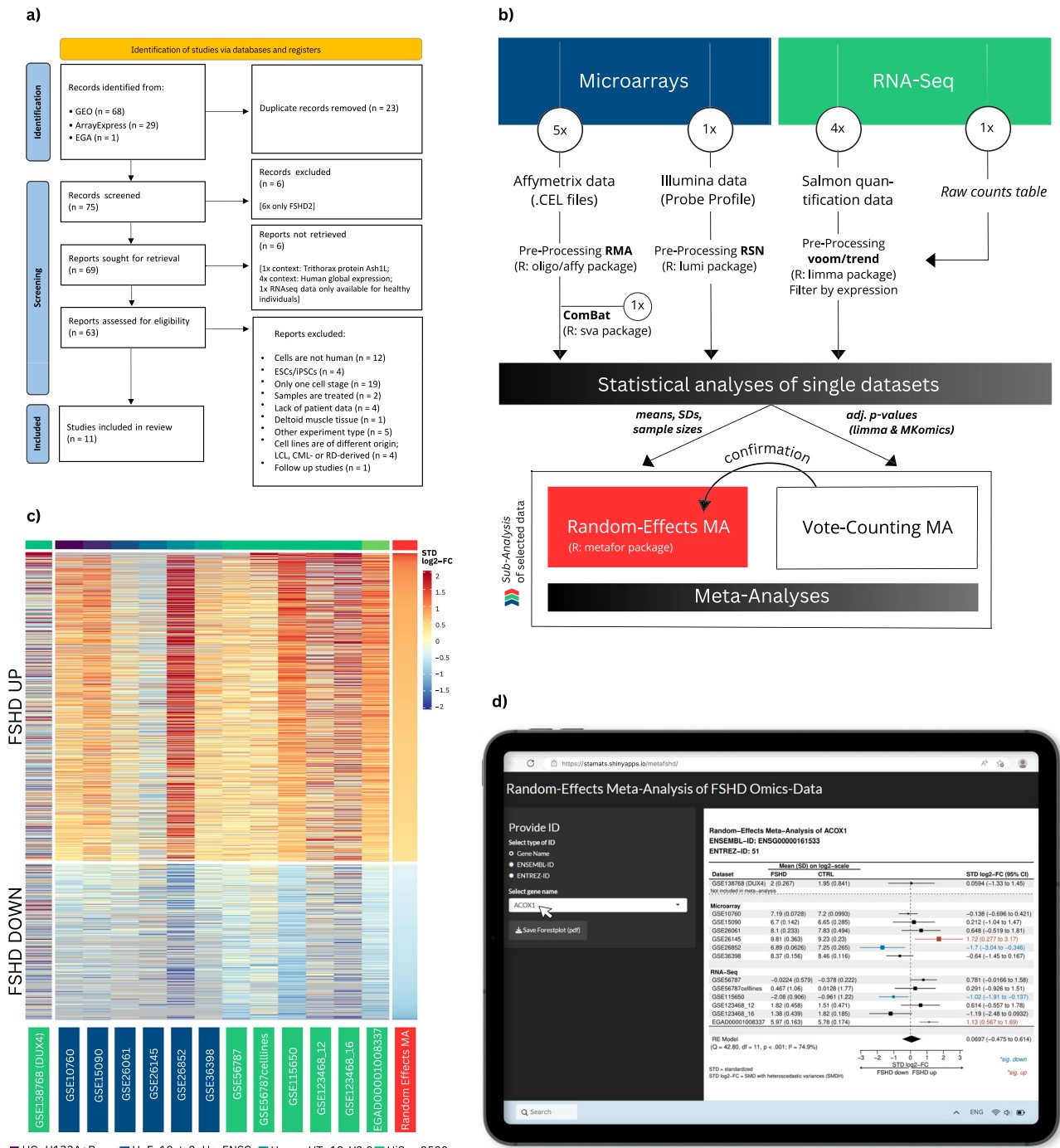

**Fig. 1 | Meta-analysis procedure and results. a** PRISMA 2020 flow diagram for new systematic reviews[14]. LCL lymphoblastoid cell line, CML chronic myeloid leukemia cell line, RD rhabdomyosarcoma. **b** Transcriptome data analysis workflow: The meta-analysis consists of both microarray (blue) and RNA-Seq (green) datasets; it contains 5 Affymetrix (as CEL files), 1 Illumina microarray (as bead summary data), 4 RNA-Seq (quantified via Salmon), and 1 RNA-Seq dataset for which fastq data could not be obtained for privacy reasons, but externally generated raw data (EGAD00001008337[69], shown in italics). For detailed information about the data analysis, we refer to the section Methods and Supplementary Notes 1 and 2. **c** The heatmap shows the random-effects model results. It contains 1935 significant results (adj. *p*-value < 0.05; the exact expression is shown in Supplementary Data 1, sheet 2a). One RNA-Seq dataset contains data from an artificial DUX4-model and was

therefore excluded from the overall calculation, but was contrasted for comparison since DUX4-induced gene expression has been reported to be the major molecular signature of FSHD skeletal muscle[80]. The heatmap shows a clear separation between up- and down-regulated genes. **d** For illustration of the Meta-FSHD app, the gene *ACOX1* was chosen as an example for clarity, since it appears significantly upregulated (red) or downregulated (blue) in addition to normal expression (black), depending on the individual datasets. In the case of *ACOX1*, the diamond crosses the vertical line of no effect. Thus, the expression of *ACOX1* (STD log2-FC (95% CI) = +0.0697 (−0.475 to 0.614)) is not significantly different in the meta-analysis, with significant Cochran's *Q* test (degrees of freedom, df = 11, *p* < 0.001) and substantial heterogeneity ($I^2$ = 74.9%) between studies (Supplementary Notes 3).

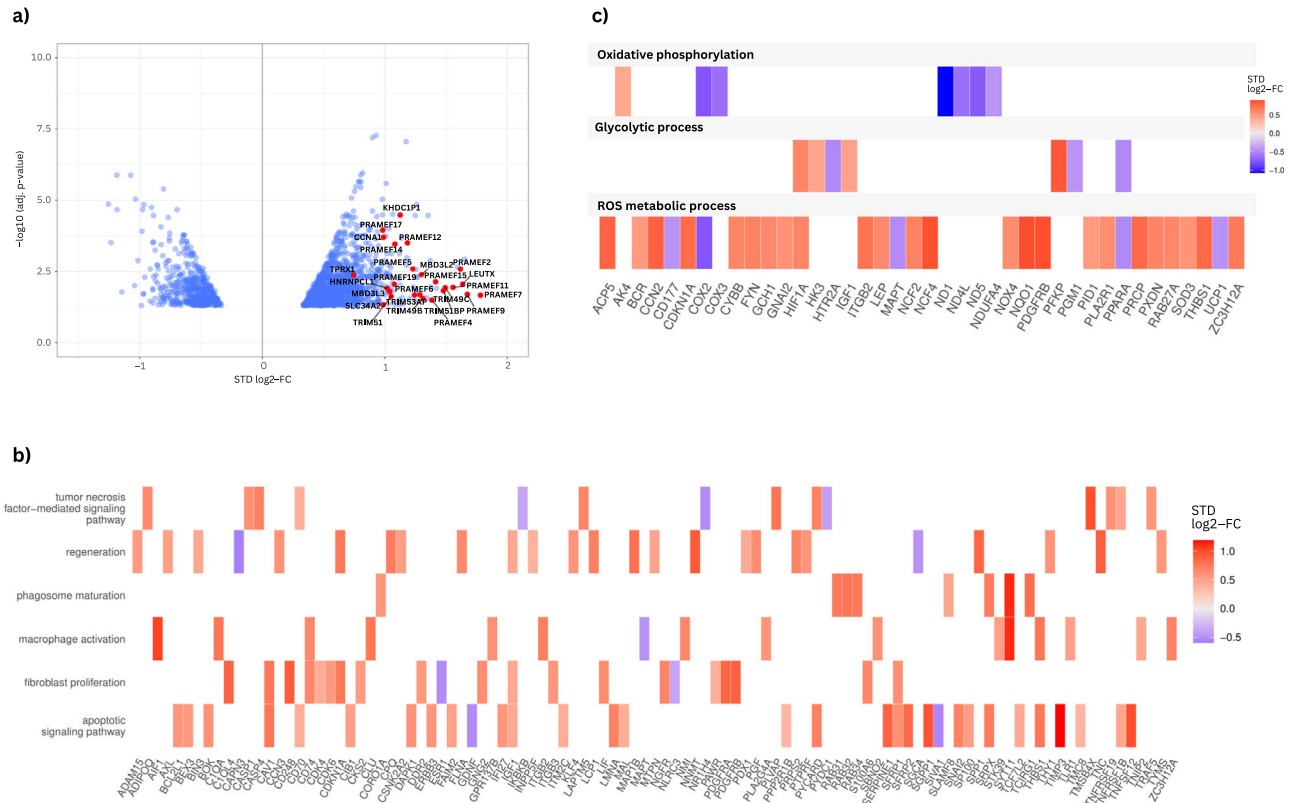

**Fig. 2 | Confirmation of previous knowledge of FSHD pathology regarding aberrant *DUX4* expression and mitochondrial impairment. a** DUX4 biomarker genes, as e.g. listed at Wang et al.[21]. are the most upregulated genes in the entire meta-analysis (Supplementary Data 1, sheet 2a). **b** The meta-analysis data show strong upregulation of genes associated with inflammasome, fibrosis, and apoptotic processes; genes linked to regeneration are also upregulated. **c** OXPHOS genes are among the most downregulated genes. Looking at the gene list of significantly differently expressed genes between patients and controls (Supplementary Data 1, sheet 2a), metabolism-related genes are dysregulated. While genes involved in reactive oxygen species and glycolytic processes are generally upregulated, OXPHOS genes at complexes I and IV are downregulated. This refers to genes regulated by the mitochondrial genome (*COX2* and *COX3* in complex IV and *ND1*, *ND4L*, and *ND5* in complex I) but also to other genes, such as *NDUFA4*, associated with both complex I and complex IV[30].

+0.619). The meta-analysis also shows the downregulation of respiratory chain genes. In complex IV, *COX2* (which is among the approximately 5% of the most downregulated genes; STD log2-FC (95% CI) = −0.759) and *COX3* (STD log2-FC (95% CI) = −0.622), two of the three mitochondrial DNA-derived genes involved in cytochrome c oxidase activity[30], were strongly downregulated. In complex I the downregulation refers to *ND4L* (STD log2-FC (95% CI) = −0.625), *ND5* (STD log2-FC (95% CI) = −0.736) and *ND1* (STD log2-FC (95% CI) = −1.07). The latter constitute three of the seven subunits of NADH dehydrogenase that originate in mitochondrial DNA and catalyze the electron transfer of NADH through the respiratory chain[30]. Intriguingly, *ND1* is among the 10 most downregulated genes of all 1935 significant genes (Fig. 2c; Supplementary Data 1, sheet 2a). The GO terms in the enrichment analysis highlight the relevance of the findings on dysfunctional OXPHOS in FSHD. Of the first 25 downregulated cellular component (CC) categories with comparatively smallest *p*-values, 10 refer to the mitochondrial respirasome, either complex I, IV, or both (Supplementary Data 1, sheet 3).

Another area that emerges from the meta-analysis is the impact of the nuclear lamina (NL) on FSHD pathology, as already reported in several studies[31,32]. This is highly interesting in terms of genome regulation, particularly with regard to the question of why skeletal muscles are so severely affected in FSHD. Our list of significant genes (Supplementary Data 1, sheet 2a) supports the findings on long-distance interactions between D4Z4, the NL, and the telomere[32], showing altered expression of *FAT1* (STD log2-FC (95% CI) = +0.576) and *SORBS2* (STD log2-FC (95% CI) = +0.583). Interestingly, *FAT1*, reported previously to be lower in FSHD muscles compared to control muscles[33], is downregulated in the DUX4-model but upregulated in almost all patient datasets. As a general trend, we find genes

associated with the NL expressed in opposite directions when comparing the DUX4-model with the patient datasets. This not only refers to long-distance interactions but also to genes directly involved in the scaffold of the NL, like *LMNA* (STD log2-FC (95% CI) = +0.698), which has already been associated with several muscle diseases[34]. The differential expression of NL-associated genes in the DUX4-model and the patient datasets suggests mechanisms independent of *DUX4*, as previously shown concerning *FAT1*[33]. These findings show altered genome organization, evident from our enrichment analysis regarding the first entry of upregulated BP pathways, which concerns supramolecular fiber organization (GO:0097435; *p*-value: 8.592309e[-09]; Supplementary Data 1, sheet 3). It encompasses a total of 102 genes (including *FAT1* and *SORBS2*), which are differently regulated between FSHD patients and controls in the meta-analysis.

## Discoveries on FSHD

Using the simplifyEnrichment package for GO clustering[35], we became aware of genes within the CC category (in both up- and downregulated pathways) that are involved in pre- and postsynaptic processes, membranes and transitions, and neuronal processes of nerve projection (Supplementary Fig. 2). Interestingly, these neuronal aspects have not been described in FSHD, yet. To test the relevance of these findings, we examined our enrichment analysis results (Supplementary Data 1, sheet 3) and detected that upregulated signaling pathways with small *p*-values refer to processes within the extracellular matrix (ECM; Fig. 3d). Since muscle fibers are located within the ECM in a three-dimensional scaffold composed of various collagens, glycoproteins, proteoglycans, and elastin, the ECM is vital for muscle contraction, integrity, and elasticity[36]. Notably, overgrowth of the ECM, also referred to as fibrosis, is well described in FSHD and results in

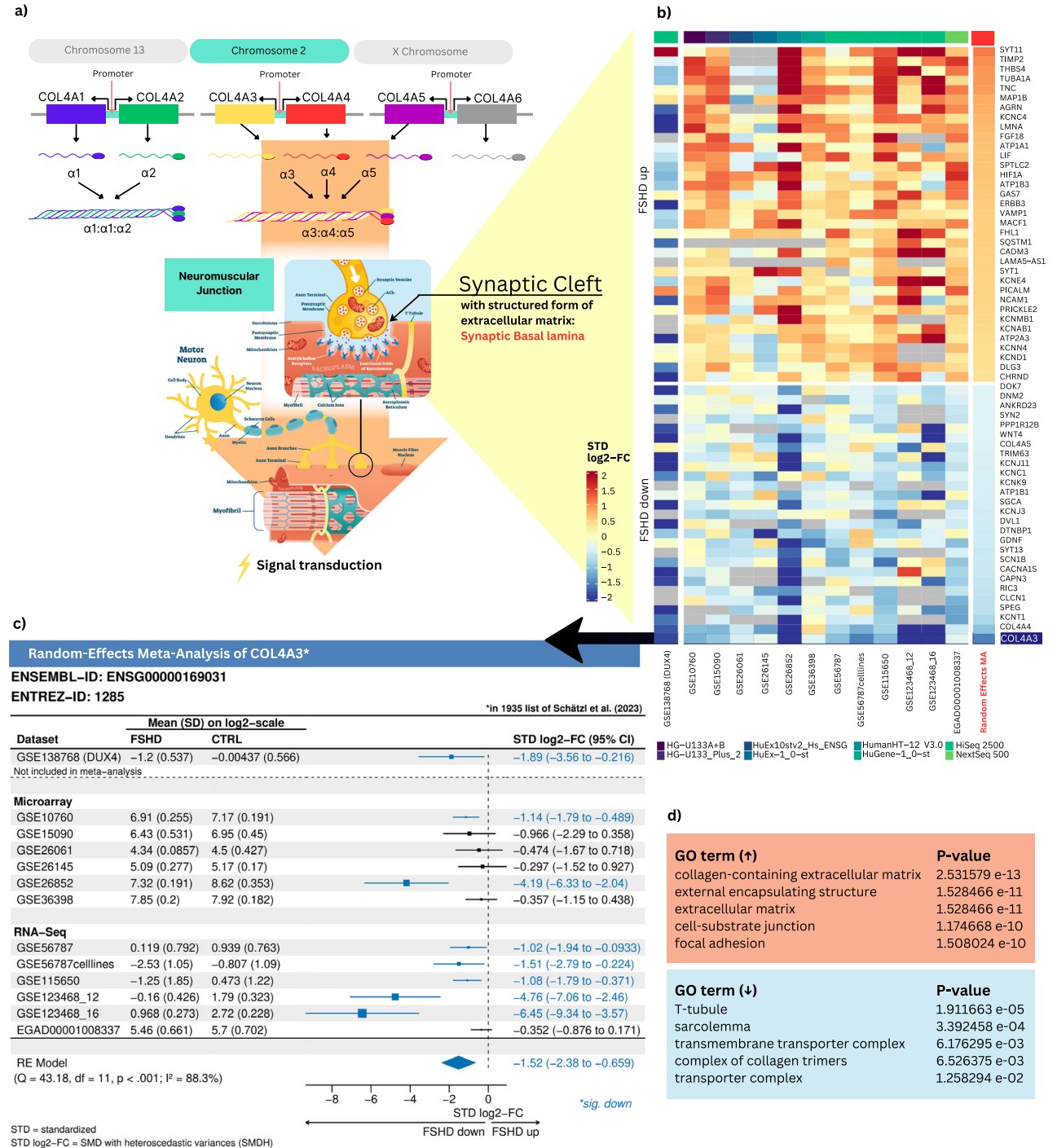

**Fig. 3 | Alterations at the NMJ and their possible influence on signal transduction. a** Synaptic BL-specific isoforms of type IV collagen genes and their possible influence on signal transduction at the NMJ: The six collagen IV genes (*COL4A1–COL4A6*), located head-to-head in pairs on three different chromosomes, generate six different α-chains. Only three groups of triple helical molecules have been detected in vivo. These are α1:α1:α2, α3:α4:α5, and α5:α5:α6. Almost all basement membranes contain the subtype with the chain composition of α1:α1:α2. In this regard, *COLA4A1* and *COL4A2* are both upregulated in the meta-analysis. *COL4A6* was not significantly differentially expressed. Interestingly, the α3:α4:α5 subtype is localized in the basement membrane of the neuromuscular synapse among others[38,39,120], raising the hypothesis of impaired signal transduction in FSHD. **b** The random-effects meta-analysis shows that, in addition to the synaptic BL-associated genes *COL4A3, COL4A4*, and *COL4A5*[38], many other genes, which have previously been associated with the NMJ, are differentially expressed between FSHD patients and controls (such as *AGRN, MACF1, DOK7, WNT4, DVL1*, etc.; Supplementary Table 3). Interestingly, most of the upregulated genes are expressed in the opposite direction in the DUX4-model. For the downregulated genes, the direction of expression is relatively similar. **c** As illustrated in the meta-FSHD app, *COL4A3* is the most downregulated significantly differentially expressed gene in the meta-analysis with STD log2-FC (95% CI) = −1.52 (−2.38 to −0.659). **d** Top 5 Enrichment analyses results on CC pathways: While ECM genes are upregulated, downregulated pathways (consistent with the information on downregulation of type IV collagen isoforms α3, α4, and α5; Supplementary Data 1, sheet 2a) indicate problems at the NMJ. Further information (gene numbers and fold enrichments) can be found in Supplementary Data 1, sheet 3 for CC pathways.

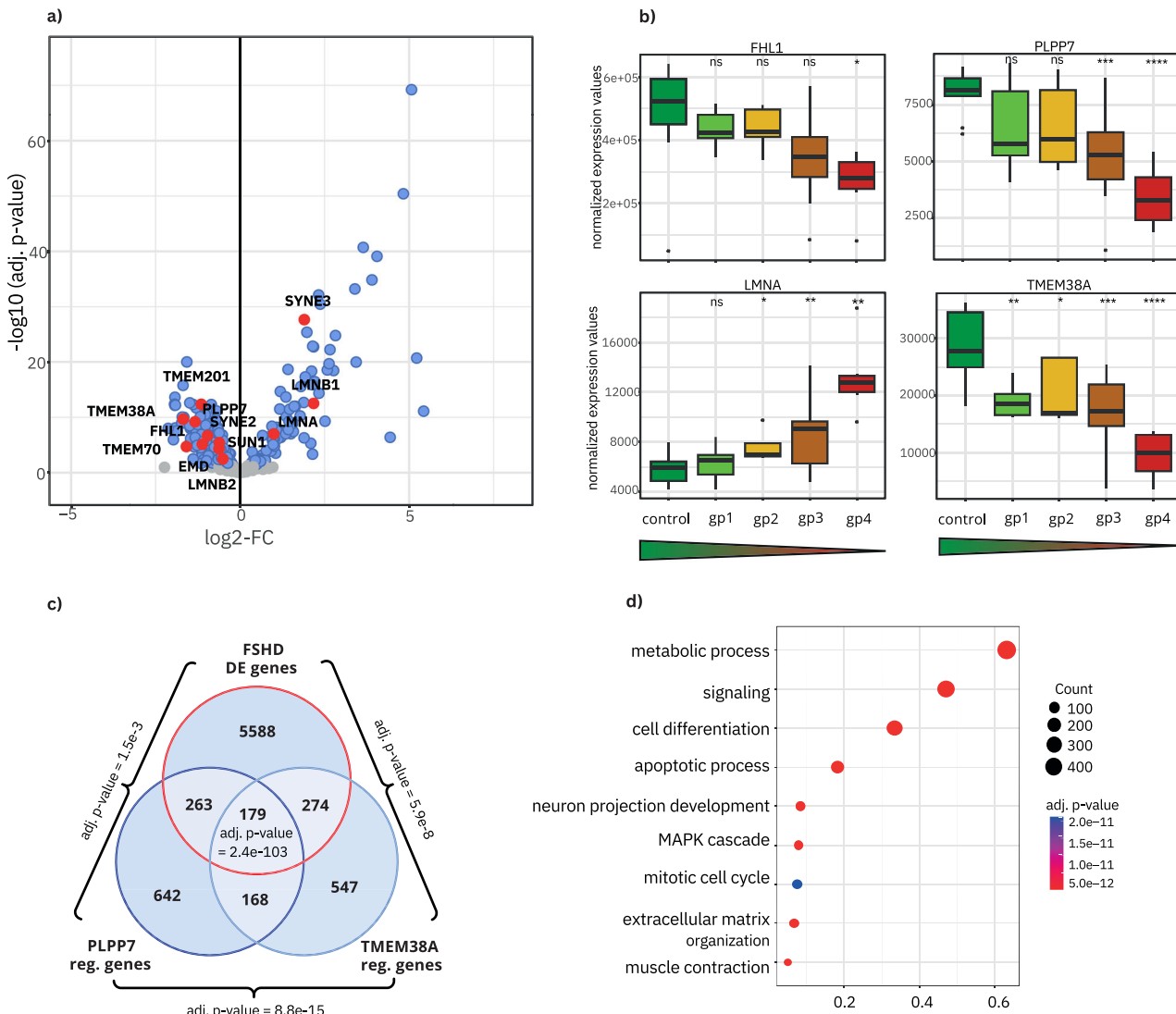

**Fig. 4 | Genes encoding NE proteins are misregulated in FSHD and contribute to the phenotype. a** Expression changes of NE genes in strongly affected FSHD patients (group 4) compared to controls. No change in grey, change in blue (log2-FC > 0.5) and muscle disease-related genes in red. The majority of NE-associated genes is downregulated. **b** Expression of four disease-related NE-associated genes in the five groups (patients vs. controls) sorted after disease severity (controls in green, group 1 light green, group 2 orange, group 3 dark orange, group 4 red). Significance was measured using a *t*-test with controls as the reference group and is indicated by asterisks (ns = not significant; *adj. *p*-value < 0.05; **adj. *p*-value < 0.01; ***adj. *p*-value < 0.001; ****adj. *p*-value < 0.0001). *FHL1*, *TMEM38A*, and *PLPP7* expression are inversely correlated to disease severity, while *LMNA* expression increases with severity. **c** Differentially expressed genes in FSHD (group 4) were compared with genes regulated by TMEM38A and PLPP7, with 716 genes potentially under their control. Adj. *p*-values (Holm's method) have been calculated using the R package SuperExactTest[121]. **d** A GO term enrichment analysis of NET regulated genes shows that 716 genes fall in GO terms relevant to the FSHD pathology.

hardening of the interconnective tissue, which leads to impaired muscle contraction and stiffness[37]. Consistently, our data also show many genes upregulated at the deepest and smallest component of the ECM, the basal lamina (BL), which is adjacent to the sarcolemma of the myofibers (such as type IV collagen genes, which are reported to dominate the BL (upregulation of *COL4A1* and *COL4A2*), specific laminins (*LAMA2*, *LAMB1*, *LAMA5-AS1*), several integrins and elastin[38]. Due to this general upregulation of ECM-related genes, we were surprised to find one very specific set of collagens downregulated, which is reported to form a functional trimer at the NMJ (Fig. 3d). Intriguingly, the BL differs in morphology depending on whether it is synaptic or extra-synaptic. The synaptic BL is composed mainly of type IV collagen α3-, -α4-, and -α5-chains (Fig. 3a), which, together with certain laminins (laminin β2, α4 and -α5) serve transmission of signals and mechanical forces that perform muscle innervation[38–40]. Notably, all three synaptic type IV collagen isoforms, α3- (*COL4A3*), α4- (*COL4A4*), and -α5 (*COL4A5*), are strongly downregulated in FSHD patients compared with

controls throughout all datasets (Supplementary Data 1, sheet 2a). *COL4A3* is even the most downregulated gene across the entire meta-analysis (STD log2-FC (95% CI) = −1.52 (−2.38 to −0.659), Fig. 3c). We thus hypothesised that NMJ architecture may be altered in FSHD patients probably affecting muscle innervation and searched for other factors that have been associated with impaired signal transduction at the NMJ. We discovered more than 60 genes that are misregulated on the transcriptional level (including *AGRN*, *MACF1*, *DOK7*, *WNT4*, *DVL1*, etc.; Fig. 3b). The data show altered processes regarding ion channels and ion pumps, NMJ maintenance and formation, acetylcholine (ACh) receptor clustering, and vesicle transfer (Supplementary Table 3).

## Nuclear envelope (NE) protein-encoding genes are misregulated in FSHD

The role of nuclear lamina-associated genes (Supplementary Data 1, sheet 2a), the alteration of supramolecular fiber organization (BP upregulation;

Supplementary Data 1, sheet 3), and CC downregulation of signaling pathways associated with the nuclear body and nuclear speckles (Supplementary Data 1, sheet 3) led us to examine the results of the meta-analysis in more detail. In this context, the meta-analysis demonstrated the variability of the FSHD phenotype, not only between patients but also between different muscle types, left and right muscles, or even different loci in the same muscle (Supplementary Note 1); besides, some FSHD samples show gene signatures like controls depending on the extraction of muscle tissue[41] (Supplementary Fig. 4). Hence, providing additional information about the degree to which the investigated muscle is affected is necessary to differentiate between actual contributors to the phenotype and noise[42]. Moreover, RNA-Seq has a clearly higher detection rate of differentially expressed genes (DEG) than microarrays and also allows the analysis of splicing[43,44]. Therefore, we used the RNA-Seq dataset generated by Wang et al.[21] using muscle biopsies of 36 FSHD patients for a deeper analysis of pathways we found altered in the meta-analysis. For the said study, Wang et al. correlated the expression of four DUX4-regulated biomarker genes with MRI data of the lower extremities and histopathological changes. Based on this, they divided the patients into four groups, with group 1 being similar to the controls and group 4 displaying the strongest pathology[21].

While the number of DEGs (log2-FC > 1 and < −1, p-value < 0.05) was already high when analyzing all samples together (1879), we found even more genes to be significantly misregulated when comparing the groups to the controls separately, with group 4 having the highest amount (7400). The reason for this could be the reduced variability within the groups, but also the increased disease pathology. Since there are so many DEGs in severely affected individuals, we wondered whether alterations in genome organization induce these changes. Several nuclear envelope transmembrane proteins (NETs), including muscle-specific ones, have been proven to be involved in genome organization and gene expression regulation[45]. Notably, a NET has been described previously to be misregulated in FSHD[46], and long-distance interactions between the D4Z4 locus and the nuclear envelope have been reported[32]. Further, there are muscular dystrophies with similar symptoms to FSHD, that are linked with genes of the nuclear envelope: striated muscle laminopathies (e.g. Emery–Dreifuss-muscular-dystrophy and limb-girdle-muscular-dystrophy 1B) are caused by mutations in *EMD*, *LMNA* or *SYNE1*, among others. We thus first screened the data for a list of 386 NE-associated genes that are known to be expressed and relevant in muscle[47,48]. Figure 4a shows the expression of these genes in strongly affected individuals (group 4) compared to controls. This revealed that many NE protein-encoding genes were significantly differentially expressed. Noteworthy, the majority were downregulated (152 down vs. 83 up with log2-FC > 0.5 and < −0.5), while the majority of all DEGs in group 4 were upregulated (2032 down vs. 5368 up). Many of these genes play a role in positioning specific genes to the NE and thereby repressing their expression[49,50]. Thus, it is conceivable that downregulation of these NE genes might contribute to the upregulation of many DEGs found in FSHD. Importantly, we found genes associated with Emery–Dreifuss-muscular-dystrophy altered in FSHD patients, including *LMNA* (2-fold upregulated) and *EMD* (1.5-fold down Fig. 4a, red). We then checked whether the expression of these genes correlates positively with disease severity and indeed saw a clear correlation for many of them, some of which we present in Fig. 4b. Mutations in *LMNA, FHL1, PLPP7* and *TMEM38A* have been linked to Emery–Dreifuss-muscular-dystrophy[49,51,52]. Notably, PLPP7 and TMEM38A were shown to be important for muscle regeneration as their knockdown leads to inefficient differentiation in C2C12 myoblasts. Since they are both significantly downregulated in FSHD (*TMEM38A* in all groups, *PLPP7* in groups 3 and 4), they might contribute to muscle weakness and wasting through impaired muscle regeneration. Therefore, we conjectured that an actual contribution of TMEM38A and PLPP7 to the FSHD phenotype would lead to expression changes of genes regulated by them. Employing a list of target genes of these two NETs generated in C2C12 mouse myoblasts[53], we found 716 genes misregulated in FSHD that are potentially regulated by TMEM38A and PLPP7 (Fig. 4c). When analyzing these 716 genes for a GO term analysis among others metabolism, signaling,

and differentiation were enriched in FSHD (Fig. 4d). It is noteworthy that TMEM38A and PLPP7 are only two of several genome organizing NETs being misregulated.

## Components of the splicing machinery are downregulated in FSHD and result in the mis-splicing of muscle genes

Looking at the results of the random-effects analysis, the spliceosomal complex was among the top 15 results for downregulated CC clusters (Supplementary Data 1, sheet 3), whereas downregulated mRNA splicing via the spliceosome affected almost all datasets with significant results in the vote-counting approach (Supplementary Data 1, sheet 6).

Previous studies suggested that components of alternative splicing are upregulated, while constitutive splicing is downregulated in two other muscular dystrophies, myotonic dystrophy type I and Emery–Dreifuss-muscular-dystrophy[54,55]. Since there are many similarities between muscular dystrophies also on the molecular level, we analyzed the dataset generated by Wang et al.[21] about the expression of splicing components using a gene set enrichment analysis for splicing associated terms. While splicing factors were misregulated in all samples, groups 2 and 3 (Supplementary Figs. 5 and 6) showed upregulation of alternative splicing and downregulation of constitutive splicing. In contrast, there was a general downregulation of alternative as well as constitutive splicing in group 4 (Fig. 5a). This suggests a major disruption of the splicing machinery in strongly affected FSHD patients. We hypothesize that fewer spliceosomes assemble at splice sites due to the lower abundance of splicing factors, leading to many (constitutive and alternative) splice sites being unused. This has to be experimentally validated in the future but is beyond the scope of this analysis. We next looked into splicing variations, primarily in group 4, using modeling alternative junction inclusion quantification (MAJIQ)[56] to identify differentially used splicing events. We set the default of 10% (percent spliced in, Ψ-value ≥ 0.1) and false discovery rate of 10% for local splicing variations to be significantly different. We found 730 events differentially used between controls and FSHD patients (Supplementary Data 1, sheet 7). In line with a general downregulation of splicing components, we found many exon skipping events in FSHD patients (Fig. 5b). However, we were startled to detect a similarly large number of introns retained in controls but spliced out in FSHD patients. If a downregulation of many splicing factors actually leads to fewer spliceosomes assembling to functional units, we would expect less splicing in general. We thus checked these intron exclusion events separately. We found that only around 25% of these events are classic intron splice events, while 75% are exon skipping events naturally excluding the adjacent intron which was retained at a higher ratio in controls but removed along with the exon in FSHD patients. MAJIQ correctly identifies these events as introns retained in controls with a Ψ-value ≥ 0.1. In this regard, nonsense-mediated-decay (NMD) might play a role as it has previously been shown to be downregulated in FSHD and intron retention leads mostly to mRNA-NMD[57,58]. This is an important regulatory level in gene expression governed through alternative splicing. In this context, the meta-analysis shows a high amount of significantly dysregulated RNA isoforms (Supplementary Data 1, sheet 2a), which might be possible substrates for NMD. Besides, a key player of NMD, the exon junction complex component EIF4A3, which has been previously discussed by Shadle et al.[57], is significantly upregulated in the meta-analysis results (STD log2-FC (95% CI) = +0.651) and could be an important candidate for further research.

To investigate the potential effect on the muscle phenotype, we next looked into the genes mis-spliced in FSHD. A GO term enrichment analysis revealed that these genes are highly relevant for muscle-specific signaling, development, and innervation (neuron projection morphogenesis; Fig. 5c). The heatmap in Fig. 5d shows a selection of genes differentially spliced in FSHD, many of which are involved in pathways we found misregulated in our meta-analysis, e.g. in muscle structure, mitochondria and metabolism, signaling, and splicing. Interestingly, microtubule-associated factor 1 (MACF1), which is a top hit in the meta-analysis (Supplementary Data 1, sheets 2 and 4) and regulates myonuclear positioning at the NMJ[59], shows a preference for skipping of exon 116 in FSHD (Ψ-value 0.325). MAJIQ

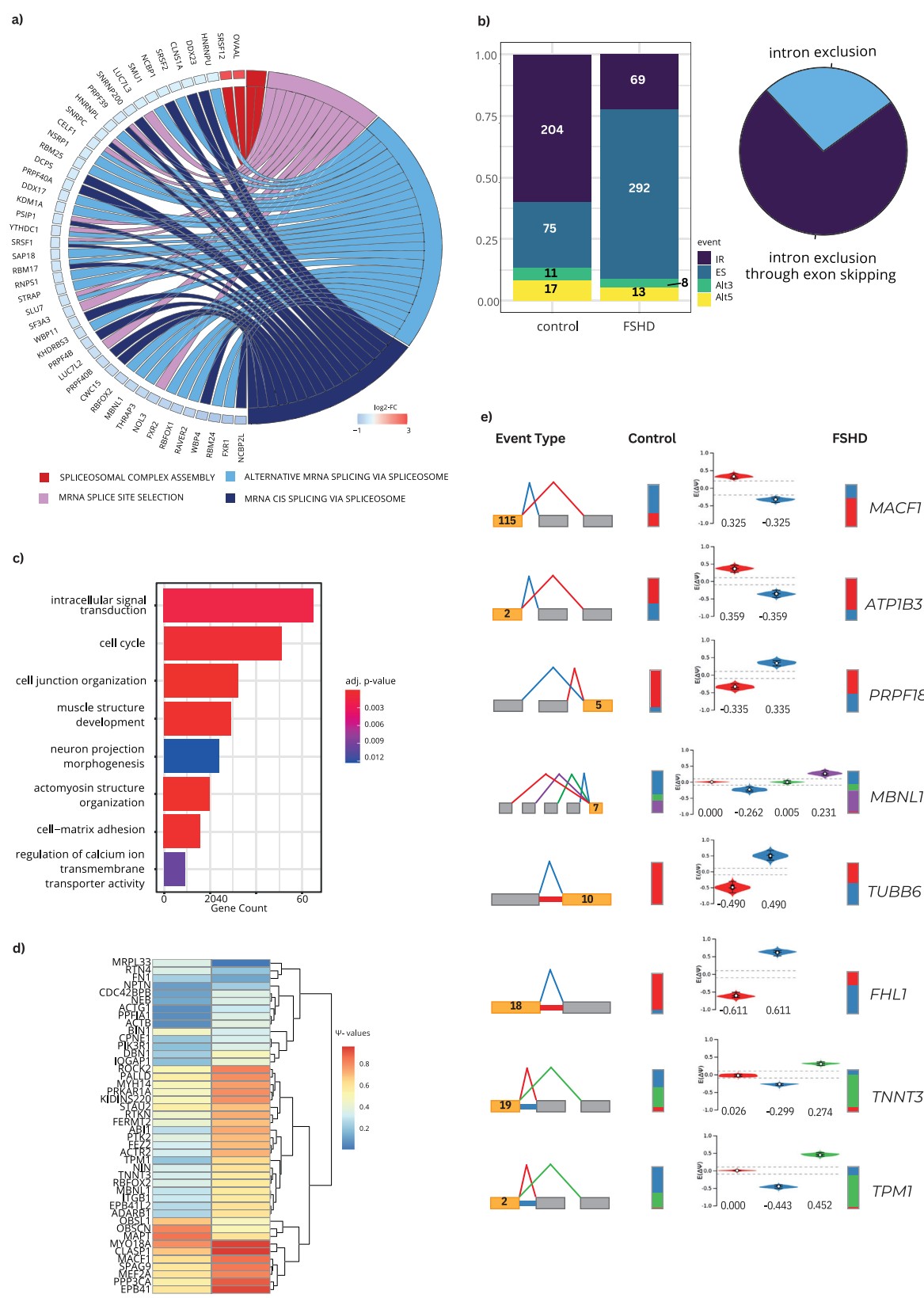

further detects an exon skipping event in *ATP1B3* ($\Psi$-value = 0.359), which encodes a subunit of the Na+/K+-ATPase and is involved in the electrical excitability of muscle and nerves. We further identified several splicing factors alternatively spliced. Interestingly, *MBNL1*, a main contributor to myotonic dystrophy, shows the same double exon skipping event of exons 5

and 6 as in myotonic dystrophy type I. There are intron exclusion events in β-tubulin *TUBB6* and *FHL1*, of which the latter is a component of the nuclear envelope and differentially expressed in FSHD, as described above. TUBB6 was proposed to act in muscle regeneration and be upregulated in dystrophy muscle[60]. We also show two examples of intron retention events

**Fig. 5 | Spliceosome components are downregulated in strongly affected FSHD patients (group 4). a** Splicing factors and splicing regulators are downregulated in FSHD. **b** The downregulation of spliceosome components leads to increased exon skipping (ES) events in FSHD. In controls, many intron retentions (IR) events are found that are not present in FSHD patients. However, ~75% of these intron exclusion events in FSHD have actually skipped exons, which naturally include adjacent introns. Around 25% of introns are spliced out although retained in controls, while there are also intron retention events in FSHD patients that are not present in controls (pie chart). The amount of alternative splice site usage (3' Alt3, 5' Alt5) is slightly reduced in FSHD patients. **c** All genes with differentially used splicing events were used for a GO term enrichment analysis, showing that these genes contribute to the known FSHD phenotype. **d** Heatmap of selected genes with alternative splice site usage; colour indicates Ψ-values (percent spliced in), and all events have at least an absolute ΔΨ-value of 0.1 between controls and FSHD patients. **e** Exon skipping (ES) and intron retention (IR) events are differentially used in FSHD patients. Reference exons are indicated in yellow and colored lines show the possible local splicing variations. Bars show proportional usage of event usage in controls and FSHD patients; violin plots display Ψ-values of the respective local splicing variations.

in controls instead of exon skipping in FSHD patients in two highly important sarcomeric genes, skeletal muscle troponin 3 (*TNNT3*) and tropomyosin 1 (*TPM1*).

## Discussion

In the past, several studies have been conducted using mainly explorative meta-analytical approaches that yielded highly interesting results. For example, a link between b-catenin and *DUX4* was discovered[61] and PAX7 target genes were shown to be globally repressed in FSHD skeletal muscle[62]. Furthermore, skeletal muscle regeneration in FSHD was found to correlate with disease severity[63]. A recently published study also showed that FSHD muscle exhibits disruption of fibroadipogenic progenitor cells, mitochondrial function and alternative splicing independent of inflammation[64]. The latter underpins the results of our meta-analysis and shows the great importance of these aspects for the disease course in FSHD.

This meta-analysis confirms *DUX4* as a main driver of FSHD pathology, as shown by the upregulation of DUX4 biomarker genes. However, many genes behave seemingly independently of *DUX4* expression. Since disease severity is sometimes comparatively mild or severe regardless of the number of D4Z4 repeats, these genes may be disease modulators under a *DUX4*-independent mechanism. This is also suggested by other research groups and highlights the need for new biomarkers to track disease progression and stratify patients[65]. Based on the meta-analysis results, we hypothesize that gene expression alterations affecting the NMJ, NE and spliceosome are relevant factors in FSHD disease progression.

We identified downregulation of genes essential for the NMJ architecture. A consequence is very likely impairment of muscle innervation in FSHD. This might be the key to a highly interesting area of research that addresses specific force in FSHD. Intrinsic force production capacity of muscle (force per unit area; N/cm²) was found to be decreased in patients with both mild and severe FSHD, regardless of disease severity and even before the onset of fat infiltration or lower limb weakness[66]. While possible reasons for these observations have been associated with early myopathic changes and non-muscular factors such as fatigue or musculoskeletal pain, our results are showing that impairment of muscle innervation may play an essential role in FSHD pathology.

Moreover, researchers have suggested that the epigenetic landscape is the missing link between the FSHD phenotype and underlying genetic parameters[67]. Besides, the major effect of DNA methylation on 3D genome structure has been described previously[68]. In this context, the altered expression of NE proteins in the meta-analysis likely contributes to the high amount of differentially expressed genes in FSHD. NE proteins are involved in genome organization and subsequent gene expression control[45]. Therefore, the consequence of the misregulation of NE proteins would be a secondary effect on many other genes. While this is difficult to filter out in multifactorial disease, there is clear evidence of this being true as genes that have been shown under the control of the muscle-specific NETs PLPP7 and TMEM38a (in mouse myotubes[53]) are also misregulated in FSHD, where the expression of *PLPP7* and *TMEM38A* appears to be correlated to disease severity. Similar effects have been observed in myotonic dystrophy type I[54]. Thus, it may reflect a more general pathomechanism in muscular dystrophies. However, due to the correlation with disease severity, the expression of NE genes may be predictive, which should be further investigated.

As an additional contributing factor, we identified altered expression of genes encoding spliceosome proteins and, subsequently, a different splicing profile affecting a bulk of genes with essential roles in muscle. In the most severely affected patients, we hypothesize that there are fewer functional spliceosomes: it seems they assemble only at every second or third splice site compared to controls, as we also observe many double exon skipping events.

Another, although already described factor, are mitochondrial impairments in FSHD[27]. Given the confirmation of these defects by this meta-analysis, studies specifically testing mitochondria-targeted agents, NAD+ precursors, or OXPHOS modulators to slow disease progression should be endorsed.

Although we lack the biopsy material to validate all these aspects of the FSHD pathology, we could identify secondary effects on general gene expression and splicing, most likely caused by the NE and spliceosome alterations. Thus, we rate them and the NMJ alterations, as potential candidates involved in the FSHD phenotype and strongly endorse a deeper investigation.

## Methods

### Individual participant data (IPD) meta-analysis

The key advantage of our meta-analysis and the foundation of its statistical power is using the original omics data from the included studies instead of summarized data. All datasets, which were generated with different approaches and at different time points, could thus be standardized in terms of data retrieval from databases and uniform analysis procedures to provide optimal conditions for directly comparing significantly differentially expressed genes and molecular signalling pathways between FSHD patients and controls.

The meta-analysis follows the preferred reporting items for systematic reviews and meta-analyses (PRISMA) statement[14] and the guidelines described by Ramasamy et al.[15]. Timing and decision-making within the meta-analysis were divided into the following work areas: Literature search, inclusion and exclusion criteria, data extraction, quality assessment, data preprocessing, statistical analysis, and data synthesis (Supplementary Notes 1 and 2, Supplementary Table 1 and Supplementary Data 1, sheet 1). In May 2022, the meta-analysis was registered in the International Prospective Register of Systematic Reviews (PROSPERO; ID: CRD42022330489[13]). As shown in Fig. 1a, our search identified a total of 11 studies that met the predefined criteria based on data from databases[69–79], related publications and information obtained during our literature search through email contact with the corresponding authors (detailed description in Supplementary Notes 1 and Supplementary Data 1, sheet 1).

In total, the meta-analysis comprises data from 300 samples, including 171 FSHD samples and 129 controls (Table 1, based on Supplementary Table 1). 13 datasets were generated from the 11 studies, as one dataset included both cell lines and biopsies (GSE56787[76]). Furthermore, one dataset included two families (two sisters each as patient and control), which resulted in family-related batch effects in our statistical analysis due to strong genetic similarity in the families (GSE123468[77]). The DUX4-model found in our literature search was not included in the overall calculation due to its artificial nature, but was contrasted for comparison since DUX4-induced gene expression has been reported to be the major molecular signature in FSHD skeletal muscle[80].

## Statistics and reproducibility

All codes for preprocessing and reproducible analysis are published on GitHub (https://github.com/FSHDresearch/Meta-Analysis-of-FSHD). The meta-analysis encompasses data from five Affymetrix GeneChips[81], one Illumina BeadArray[82] and five Illumina RNA-Seq studies[83]. All downstream analyses were performed with R (v4.2.2)[84]. For statistical analysis, the Benjamini–Hochberg correction was used to adjust for multiple testing[85] and adj. p-values of <0.05 were considered significant. All statistical tests were two-sided.

In case of RNA-Seq data, Salmon (v1.9.0) was used to quantify transcript abundance from fastq raw files (which were freely available in the databases except for EGAD00001008337), since it has been shown to significantly improve the accuracy of abundance estimates and the sensitivity of subsequent differential expression analysis[86]. In case of the EGA dataset, after email request we were provided with the raw counts table and the metadata (Supplementary Notes 1), which were subsequently integrated into the unified analysis procedure of the meta-analysis. In order not to lose data, no separate filtering steps were applied in the R workflows for the individual datasets. For pre-processing Affymetrix microarray data, the robust multi-array average (RMA) approach[87] of the Bioconductor packages oligo (v1.60.0)[88] and affy (v1.74.0)[89] were used; and for Illumina microarray data we selected Robust Spline Normalization (RSN)[90] of the lumi package (v2.48.0)[91]. As we used limma (v3.52.2)[92] in combination with MKomics (v0.7)[93] for differential expression analysis of microarray data[94], for reasons of comparability, we chose limma for RNA-Seq data preparation and analysis as well. Thus, only minimal pipeline changes were required to switch between analyses for RNA-Seq and microarray experiments[95,96].

As a first strategy a random-effects model was selected to identify significantly differentially expressed genes between patients and controls[16]. In this context, we used SMDH as an effect measure, as suggested by Bonett (2009)[17]. The model was chosen because, besides the biological heterogeneity between study participants, a relevant degree of technical heterogeneity could be assumed due to the use of different technologies (microarray vs. RNA-Seq; Supplementary Notes 3 and corresponding sensitivity analyses in Supplementary Notes 4).

Without the DUX4-dataset, we identified a total of 53113 unique Ensembl IDs (Ensembl database version 108 from December 2022). We decided to use only unique IDs measured in at least three datasets to get reliable results from the meta-analyses, which gave us 26,858 unique IDs. We filtered our data to increase the power of the analysis[97,98], whereupon 13,274 unique IDs remained. However, filtering can lead to a bias in the false discovery rate calculation[99]. The results in Supplementary Notes 7 suggest a true false discovery rate of 5% to 7% for the selected 1935 genes. For comparison we have included the results of the unfiltered analysis in Supplementary Data 1, sheets 2a and 2b. Using the random-effects model, a p-value was assigned to each individual unique Ensembl ID. In this way, 13,274 associated meta-analyses were theoretically performed for each dataset within the 292 samples (without DUX4-model samples), ultimately leading to the 1935 significant results (adj. p-value < 0.05). These results are decreasingly sorted by SMDH, as shown in the heatmap in Fig. 1c, which is linked to data depicted in Supplementary Data 1, sheet 2a.

To roughly divide the functional terms related to the genes into clusters, we used the Bioconductor simplifyEnrichment package to cluster and visualize the enrichment results[35]. In this context, genes were divided into the three GO domains cellular component (CC), molecular function (MF), and biological process (BP), each distinguishing between up- and down-regulated genes (Supplementary Figs. 1–3). These clusters were used in combination with the gene list of our heatmap shown in Fig. 1c (Supplementary Data 1, sheet 2a), corresponding enrichment analyses with Bioconductor packages[100] (Supplementary Data 1, sheet 3) and the forest plots obtained with our shiny app meta-FSHD (Fig. 1d) to analyse the results of our meta-analysis in terms of existing expertise and potential new findings in the context of FSHD.

To validate the results of the random-effects model, a second alternative analysis approach, known as vote-counting[15], was performed. In this regard, the datasets were considered separately and the significant genes and molecular pathways were compared in terms of their overlap rate between datasets. 7 of the 13 datasets yielded significant results when considering the adj. p-values by using limma (v3.52.2)[92] (Supplementary Data 1, sheet 4). In this context, standard annotation packages from Bioconductor were utilized to map the data to the ENSEMBLE gene ID[101]. The core strategy was to find out, which genes are altered in FSHD. Enrichment analyses were performed using GO-database[102,103] to identify molecular signalling pathways (Supplementary Data 1, sheets 5 and 6).

## Meta-FSHD App

The goal of meta-FSHD is to support future research in FSHD. For its implementation, we used the R packages shiny[104] and shinythemes[105] in combination with the R packages metafor[106], grid[84], forestploter[107] and ggplot2[108]. The entire R code as well as the data for the app are publicly available on GitHub (https://github.com/stamats/metaFSHD). The app can easily be used by anyone without installing R and RStudio at the website hosted by Posit Software, PBC (https://stamats.shinyapps.io/metafshd). The corresponding background data for each study, including all samples, are provided in Supplementary Table 1.

In total, meta-FSHD considers all genes measured in at least 3 datasets, corresponding to 26,858 unique ENSEMBL-IDs, 21,080 unique ENTREZ-IDs and 22,791 unique gene names. In this regard, the user can choose between gene name, ENSEMBL-ID, and ENTREZ-ID to enter the corresponding gene of interest. This has the decisive advantage that, in addition to well-described genes, novel transcripts, e.g. previously described only by their ENSEMBL-ID, can be investigated. (For DUX4 and biomarker gene searches, additional information is given in Supplementary Notes 5).

Once the gene of interest is entered, a forest plot appears, encompassing all 13 datasets. Although the DUX4-model is not considered in the random-effects calculation, it is presented in addition to the patient datasets for comparison purposes. There is an additional option to save each forest plot in PDF format. The datasets are further divided into microarray and RNA-Seq datasets for clarity. The statistical calculation is based on the STD log2-FC between FSHD and control due to the different scales for microarrays and sequencing data, where STD log2-FC corresponds to SMDH[17]. The mean (standard deviation) on the log2-scale per group (FSHD or control) is given next to the name of each study, followed by a graphical representation incorporating the studies' impact (size of squares proportional to weight (inverse of standard error) of the single dataset within the meta-analysis) with the 95% CI (horizontal lines) and the corresponding numbers on the right. If a gene is significantly upregulated in FSHD compared to control samples, the app displays it in red; if it is significantly downregulated, it is shown in blue. If a gene is insignificant, the CI-line crosses the vertical line of no effect (STD log2-FC = 0). The overall result of the meta-analysis is represented by the diamond. Furthermore, there are additional parameters such as the Cochran's $Q$ test for heterogeneity[109] and $I^2$, a measure of heterogeneity (Supplementary Note 3).

## Sub-analysis of an MRI-informed RNA-Seq dataset

Raw data from Wang et al.[21] were mapped to the human genome assembly GRCh38 (hg38) and sorted by coordinate using STAR 2.7.9a[110] for analysis in DESeq2[111] and MAJIQ[56].

A gene count matrix was generated applying featureCounts[112] and standard DESeq2 workflow was followed, inbuilt lfcShrink function was used with apeglm[113]. Patients were grouped according to the authors' assessment based on biomarker expression (LEUTX, KHDCL1, TRIM43, and PRAMEF2), which correlated with pathology (section stainings and MRI). Thus, five groups were formed: controls and FSHD groups 1–4, with the latter having the strongest phenotype and biomarker expression. All DESeq2 results are found in Supplementary Data 1, sheet 8.

A comprehensive list of 386 genes that are associated with the nuclear envelope (either transmembrane proteins or interacting with them on the nucleoplasmic or cytoplasmic side) and preferentially expressed in muscle[47,48] (Supplementary Data 1, sheet 9) was used to screen for nuclear

envelope genes misregulated in FSHD. This was done for either group 4 or all groups and differences were evaluated by calculating p-values with ggpmisc[114]. Genes regulated by TMEM38A and PLPP7 were extracted from Robson et al.[53], and the venn diagram was generated with ggVennDiagram[115]. GO term enrichment analysis (for NET regulated genes and mis-spliced genes) was conducted using gprofiler2[116] and visualized with enrichplot[117]. Next to classic GO term analysis, gene set enrichment analysis was used for finding splicing-related pathways and their genes with the Bioconductor package fgsea[118], which was then visualized using GOplot[119].

Splicing analysis was done with MAJIQ (v2.3) in python (v3.9; https://www.python.org/) for group 4 compared to controls and visualized using Voila (v2.3; https://majiq.biociphers.org/). All other plots were generated with ggplot2[108].

## Reporting summary

Further information on research design is available in the Nature Portfolio Reporting Summary linked to this article.

## Data availability

The raw data are from Gene Expression Omnibus (GEO). One RNA-Seq dataset, for which no fastq data were available for data protection reasons, is from The European Genome-phenome Archive (EGA). For the latter we obtained raw counts table and the metadata. Detailed information on the data used in the meta-analysis is provided in Table 1 and Supplementary Notes 1 and Supplementary Table 1. Complete results of the analyses are included in Supplementary Data 1.

## Code availability

The complete R code of the meta-analysis is available on GitHub (https://github.com/FSHDresearch/Meta-Analysis-of-FSHD). The entire R code as well as the data for the meta-FSHD app are publicly available on GitHub (https://github.com/stamats/metaFSHD). The app can easily be used by anyone without installing R and RStudio at the website hosted by Posit Software, PBC (https://stamats.shinyapps.io/metafshd).

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

## Acknowledgements

We would like to thank three reviewers and the editors for their critical reading of our manuscript and their helpful suggestions. We acknowledge financial support by DFG within the funding programme Open Access Publikationskosten.

## Author contributions

T.S.: conceptualisation, methodology, data curation, formal analysis, investigation, visualisation, writing—original draft preparation, writing—review & editing; V.T.: conceptualisation, data curation, formal analysis, investigation, visualisation, writing—original draft preparation, writing—review & editing; L.K.: conceptualisation, investigation, writing—review & editing; H.W.: data curation, investigation, writing—review & editing; B.S.: conceptualisation, investigation, writing—review & editing; H.-P. D.: conceptualisation, investigation, writing—review & editing; P.M.: supervision, conceptualisation, methodology, formal analysis, investigation, visualisation, writing—original draft preparation, writing—review & editing; M.K.: project management, supervision, conceptualisation, methodology, data curation, formal analysis, app implementation, investigation, visualisation, writing—original draft preparation, writing—review & editing; All authors approved the final version of the paper for submission.

## Funding

## Competing interests

The authors declare no competing interests.
