## [Peer review file · Communications Biology]

Reviewers' comments:

Reviewer #1 (Remarks to the Author):

Schätzl et al. present a meta-analysis of gene expression data for FSHD patients and controls. It is a useful resource for the FSHD research community, made very user friendly with a publicly available Shiny App. Apart from a methodological point to carefully check, I only have minor suggestions to improve clarity of the main text and figures.

Major:

- filtering. The supplementary methods suggest that genes are first filtered on log-fold change (greater than 0.23), and then the remaining genes are tested (for a significant log-fold change). The filter and test statistic are not independent, which would impact the error control. Typically genes are filtered out if they have low expression (where there's not much power to find a difference anyway). Here it looks like all the bottom of the volcano plot is missing (Fig2A). The adjusted p-values and significant gene numbers are likely to change after this is rectified.

minor:

- The introduction and results lack details about the studies included, the technologies, the number of samples, the tissues evaluated, etc. It would be great to add a summarised version of Supplementary Information S3 in the main text.

- the text in figures is often too small. After the conversion of the figures into the main pdf, it's very difficult to read gene names, log-fold changes, or anything on the Fig3 schematic.

- Fig3 title is too strongly worded as a conclusion, when it's only a hypothesis based on (mild) gene expression changes.

it would be great to report gene numbers and fold enrichments for Fig3D Gene Ontology results.

- Fig4C can you calculate a p-value for the overlaps?

- Circular plots in Fig5a and supps are not very clear. A 3-column format with gene name, logFC, annotations would be a lot easier to read.

- Fig5B it would be helpful to report absolute numbers of AS events as well as their relative proportions. I wasn't sure whether this analysis was done for only one of the selected RNA-seq datasets, could you clarify? If so why not include the others?

- typos "sensitive analysis/analyses" for "sensitivity"; Fig1A has "für" instead of "for"; "enervation" for "innervation".

Reviewer #2 (Remarks to the Author):

Schatzl et al., present a meta-analysis of transcriptomes corresponding to FSHD patient muscle biopsies and cultured myoblasts and myotubes. They report mis-regulation of DUX4 target genes, metabolic, nuclear envelope, neuronal and spliceosome gene sets.

Unfortunately, the results and approach of this work are not novel, as numerous transcriptomic meta-analyses have been performed before for FSHD with similar conclusions. The manuscript also contains significant omissions and inaccuracies and in my opinion the methods are flawed.

Major omissions and inaccuracies:

A PubMed search for 'FSHD transcriptomic meta-analysis' reveals 4 publications dating back 8 years:

- Banerji et al., b-catenin is central to DUX4 driven network rewiring in facioscapulohumeral muscular dystrophy, *Journal of The Royal Society Interface*, 12(102):20140797, (2015).
- Banerji et al., PAX7 target genes are globally repressed in FSHD skeletal muscle, *Nature Communications*, 8(2152) (2017).
- Banerji et al., Skeletal muscle regeneration in Facioscapulohumeral muscular dystrophy is correlated with pathological severity, *Human Molecular Genetics*, 29(16):2746-2760, (2020).
- Engquist et al., FSHD muscle shows perturbation in fibroadipogenic progenitor cells, mitochondrial function and alternative splicing independently of inflammation. *Hum. Mol. Genet.*, ddad175, (2023).

Despite the similarity in approach and significant overlap in overall conclusions, none of this highly relevant body of literature is discussed or even cited by the authors.

A key finding of prior FSHD meta-analyses has been suppression of PAX7 target genes in patient biopsies, this is not mentioned by the authors.

The authors also incorrectly claim novelty in the title and throughout the article.

The mis-regulation of nuclear envelope genes in FSHD has been known for many years and in fact was well demonstrated by transcriptomic meta-analysis in:

Bakay et al., Nuclear envelope dystrophies show a transcriptional fingerprint suggesting disruption of Rb-MyoD pathways in muscle regeneration. *Brain* 2006 (doi: 10.1093/brain/awl023).

Schatzl et al., should be aware of this publication as it is associated with one of the data sets they considered (GSE3307), however, it is not cited or discussed in the manuscript.

The mis-regulation of the spliceosome is also well known in FSHD and has been shown in several publications over the last 8 years, for example:

Rickard et al., Endogenous DUX4 expression in FSHD myotubes is sufficient to cause cell death and disrupts RNA splicing and cell migration pathways, *Hum. Mol. Genet.*, 2015 (<https://doi.org/10.1093/hmg/ddv315>).

Other non-exhaustive specific inaccuracies include:

1. Line 164: 'Interestingly, these neuronal aspects have not been described in FSHD, yet' - This is not true mis-regulation of neuronal genes in FSHD has been known for many years, for example: Dandapat et al., Expression of the Human FSHD-Linked DUX4 Gene Induces Neurogenesis During Differentiation of Murine Embryonic Stem Cells. *Stem Cells Dev.*, 2013 (doi: 10.1089/scd.2012.0643).
2. Line 142: 'skeletal muscle is almost exclusively affected in FSHD' – This is not true, FSHD has prevalent extra-muscular manifestations, including sensorineural hearing loss (up to 30% in some studies) and retinal telangiectasia (up to 70% in some studies), this information is available in many reviews on FSHD.

Methodological Flaws:

The study selection criteria detailed in the supplementary document are contradictory in many regards and studies seem slightly cherry picked:

On genotype:

The authors state: 'Because of the slightly different pathological background of the FSHD1 and FSHD2 disease types, datasets containing solely FSHD2 samples were removed. However, individual FSHD2 samples were retained since DUX4 expression has been described as a major feature of muscle wasting in both disease types⁶⁻⁸ and some FSHD2 samples of the included studies show strong DUX4 expression'

The distinction between FSHD1 and FSHD2 is never discussed anywhere in the main manuscript, which only refers to 'FSHD' and describes only the FSHD1 genotype in the introduction. This is an important omission given that the meta-analysis does include some (but notably not all) FSHD2 samples.

The criteria for including an FSHD2 sample appears to be one of convenience, namely that the FSHD2 sample should be profiled alongside an FSHD1 sample and show high DUX4 expression. Given that a major finding of the authors is the up-regulation of DUX4 target genes, this selection criteria seems biased. One should either analyse all FSHD1 and FSHD2 samples or alternatively just select one genotype.

Moreover, in the case where both genotypes are considered there must be suitable adjustment of the differential expression (DE) analysis to account for genotype driven differences, this does not appear to have been performed.

On muscle type:

The authors state: 'In contrast, the deltoid muscle is reported to be preserved or less affected^{21,22}. Therefore, studies that solely focused on the deltoid muscle were excluded from our meta-analysis.'

The muscle involvement of FSHD is highly heterogeneous, and several muscles are spared until late in the disease process, the muscles most often associated with late involvement are deltoids and quadriceps (a summary with associated refs of muscle involvement in FSHD can be found in this review – Banerji and Zammit 2021 doi: 10.15252/emmm.202013695). To my knowledge there is no evidence that the deltoids are less involved than the quadriceps in FSHD. Hence if deltoids are excluded from the study on the basis of late involvement, we must also exclude quadriceps. Most muscle biopsies are quadriceps, so if this was done the meta-analysis would not really be possible. As with above the authors must analyse either all muscle groups, or restrict their analysis to one type (e.g., quads).

Again, if more than one muscle group is considered (as it currently is) the authors must adjust for muscle type specific gene expression in their differential expression analysis. Especially as this is well known to impact the expression of metabolic genes due to differences in slow/fast twitch fibre contributions, which are reported by the authors as altered in their FSHD meta-analysis.

In particular many of the data sets analysed contain biased and unequal muscle groups distributions between FSHD and control samples, which could drive spurious results in this analysis. Sample C6 in GSE56787, for example is the only TA sample in a data set otherwise comprising only quads and this has previously been shown to impact analysis (<https://doi.org/10.1038/s41467-017-01200-4>) – it is not clear if the authors considered this.

On mosaicism:

Immortalised cell lines derived from mosaic patients were excluded from the meta-analysis with no justification, if these were biopsies, containing a mixture of pathological and healthy tissue I would accept this decision. However, these are clonal cell lines and are the closest one can get to a perfectly controlled system for FSHD, one line has an FSHD1 mutation the sister line does not but is otherwise isogenic, excluding these lines will only reduce inference.

On myoblasts and myotubes:

As with FSHD1 and FSHD2 genotypes, studies profiling myoblasts are included when it is convenient, namely when they are profiled alongside myotubes, but otherwise excluded, without clear justification:

'However, the analysis was extended to cell line datasets when both myoblast and corresponding myotube data were available, since two main stages of myogenesis from satellite cell to fibre are

represented in these11-13. This finally allowed comparison of the respective data with those of the biopsy datasets.'

As an additional note, combining muscle biopsies and myoblast/myotube expression has many complications, which are not touched on in methods/discussion. These include immortalisation and the impact of this on telomeres/gene expression and differences in culture conditions (serum, differentiation protocol etc.).

It is well known that DUX4 is readily detected in MBs/MTs but not is almost never detected in biopsies and that these samples are very different in transcriptomic manifestation of the disease process and must be analysed separately (as has been done in previous meta-analyses).

The study GSE140261 is also omitted because it is a follow up study in a natural history of FSHD patients with no intervention, it seems odd to exclude additional data on these grounds, as they are essentially distinct technical replicates and may help minimise noise. Given that all of the cell line data sets contain technical replicates, which correspond to the same patient, and these are not excluded, why is exclusion done for the biopsy studies?

Further comments on methods:

1. The authors use limma for both microarray and RNA-seq DE analysis, this is suboptimal for RNA-seq where counts follow a negative binomial distribution a package such as DESeq2 should be used instead. Using limma will alter the distribution of significant DE genes in RNA-seq data and impact the results.
2. Micorarray and RNA-seq are very different technologies in terms of coverage and detection of the transcripts, this must be discussed in the paper as we are limited to genes represented in both.
3. The authors discuss in the SI methods about GSE123468 being split into two data sets due to batch effects caused by different families. Batch effects are due to experimental set up, not genotype, these samples were processed in the same batch and so DE analysis should be done on the whole study adjusting for genotype as a covariate, not separately on the study cut in half, the latter approach is under-powered. Conversely, there are significant batch effects in one of the studies considered GSE36398, where disease status is confounded with experimental batch, the authors do not comment on this.
4. Lines 212-216: The authors investigate the data set of Wang et al., 2019, but their findings seem confused: 'While the number of DEGs was already high when analyzing all samples together (1879), we found even more genes to be significantly misregulated when comparing the groups to the controls separately, with group 4 having the highest amount (8400). The reason for this could be the reduced variability within the groups, but also the increased disease pathology.' These 4 groups were defined by Wang et al., by how much their transcriptomes differed from control in the expression of a number of DUX4 targets, not pathological severity (that was a subsequent correlate), so it is not surprising that more DEGs are found in group 4 vs control compared to group 1 vs control, this is basically true by construction. The authors then link this expected finding of more differentially expressed genes in group 4 back to pathological severity, in a circular argument.

In light of the above omissions, inaccuracies and methodological flaws, I cannot recommend this article for publication.

Reviewer #3 (Remarks to the Author):

Schätzl et al. present a meta analysis of transcriptomic data prepared from patient-derived cell lines and / or tissue biopsies of individuals with predominantly facioscapulohumeral dystrophy (FSHD1). FSHD1 is a genetically intriguing disorder caused by an unusual non-coding, gene regulatory mechanism involving contraction of D4Z4 repeats on 4q35 and derepression of the DUX4 gene. The inverse of the D4Z4 repeat length and / or the DNA methylation status of the repeat usually, but do not always, correlate with disease severity. The aim of this study was to deeply investigate molecular pathways that are dysregulated in FSHD1 that may have been overlooked in previous studies. All data was extracted from public resources and spans 11 different studies carried out using technologies spanning 3' and exon arrays to RNA seq, between 2011 and 2022. The study is well conceived, well executed and a fantastic example of why FAIR principles as they apply to genomic data are so important. A weakness of the study is that due to the design, none of the novel findings, e.g. predicted alteration to neuromuscular junctions have been validated by orthogonal methods on patient tissue.

Major questions and comments:

1. Discussion on the involvement of dysregulated nuclear envelope proteins is confusing and requires more detail. Presumably the genes that are secondarily altered due to altered mechanotransduction are likely targets of the YAP / TAZ hippo pathway or other transcription factors which link with mechanosensitive pathways via the LINC complex. These targets may not show up when performing enrichment analyses using Biological Process, Molecular Function and Cellular Component ontologies. Further investigation of this by testing for enrichment of specific transcription factor involvement against TRANSFAC, JASPAR or similar databases would be of interest.

2. The biological explanation for the lack of available spliceosomes in the discussion is not clear. In DM1, these factors are sequestered due to expression of the CUG repeat, what is the proposed mechanism at play in FSHD, do RNA foci form in severely affected individuals with FSHD? The relationship to FSHD and disruption to nonsense mediated mRNA decay (NMD) may be of interest.

Feng Q, Snider L, Jagannathan S, Tawil R, van der Maarel SM, Tapscott SJ, et al. A feedback loop between nonsense-mediated decay and the retrogene DUX4 in facioscapulohumeral muscular dystrophy. Green R, editor. eLife. 2015 Jan 7;4:e04996.

Shadle SC, Zhong JW, Campbell AE, Conerly ML, Jagannathan S, Wong CJ, et al. DUX4-induced dsRNA and MYC mRNA stabilization activate apoptotic pathways in human cell models of facioscapulohumeral dystrophy. *PLOS Genetics*. 2017 Mar 8;13(3):e1006658.

3. In the discussion the authors comment that spliceosomes assemble at every second or third splice site. This statement needs support of functional validation, e.g. measurement of spliceosome occupancy via RIP-seq or similar. With the available data it is not possible to conclude this vs. an altered splicing of cassette exons specifically driven by dysregulation of the spliceosome. Generalised lack of splicing machinery should lead to increased intron retention generally in the transcriptome, and potentially degradation of the majority of such transcripts by NMD (as noted above this may be compromised), however data as they are presented does not support this.

Minor:

4. The manuscript reads in many places as a direct copy-paste from a thesis. The readability and clarity of the work could be improved by cutting down the volume of text and using a language style which is more direct. This is a suggestion only, I don't wish imply the author's chosen style is incorrect.

5. Page 7 lines 231 and 232: "...the majority is..." change to "... the majority were..."

Reviewers' comments

Reviewer #1

Schätzl et al. present a meta-analysis of gene expression data for FSHD patients and controls. It is a useful resource for the FSHD research community, made very user friendly with a publicly available Shiny App. Apart from a methodological point to carefully check, I only have minor suggestions to improve clarity of the main text and figures.

Major:

- filtering. The supplementary methods suggest that genes are first filtered on log-fold change (greater than 0.23), and then the remaining genes are tested (for a significant log-fold change). The filter and test statistic are not independent, which would impact the error control. Typically genes are filtered out if they have low expression (where there's not much power to find a difference anyway). Here it looks like all the bottom of the volcano plot is missing (Fig2A). The adjusted p-values and significant gene numbers are likely to change after this is rectified. We agree that any procedure that filters out features has an influence on the adjustment of the p-values. Nevertheless, filters for mean intensity, variance or fold-change are often applied in omics-data analyses with the goal to increase power. Van Iterson et al (2010) (<https://doi.org/10.1186/1471-2105-11-450>) examined these filters and concluded that the "fold change filter seems to be best, yielding considerable increases of power when used". Furthermore, we found (and still find) it impossible to establish a general mean or variance cut-off in a meta-analysis including data from different platforms. We assumed that such a general cut-off for all platforms would very likely lead to or increase platform bias in the analysis. Therefore, in our view, platform-specific cut-offs for mean intensities or variances would have to be found. But, again there would only be no additional platform bias in the analysis, if these cut-offs would be in some sense equivalent for all platforms. We are not aware of any approach for achieving this. Hence, we decided to use the standardized fold-change (SMHD) to filter the data, hoping that this filter would be as platform-independent as possible. More specifically, we filtered out all genes with an SMHD smaller than 0.2, as a Cohen's d of less than 0.2 is usually considered as a small effect (see *Supplementary Information S1*). The cut-off was selected based on the assumption that we have high heterogeneity in the datasets and that significant genes with small effects will very likely be false positives. As an additional check, we now tested all 26858 genes and looked at the (raw) p-values and SMHDs. The smallest SMHD that leads to a significant result is 0.242, so our filter does indeed seem to filter out almost exclusively null features and thus increase the power of our analysis.

Minor:

- The introduction and results lack details about the studies included, the technologies, the number of samples, the tissues evaluated, etc. It would be great to add a summarised version of *Supplementary Information S3* in the main text. We have included a shortened version of *Supplementary Information S3* as **Table 1** in the main text (line number 398 and mentions in the text: line numbers 75, 559).

Table 1: Overview of Datasets

	Author & Year	Dataset	Technology	FSHD*	CRTL
Microarrays	Cheli et al. (2011)	GSE26061	Affymetrix Human Exon 1.0 ST Array	10	6
	Tsumagari et al. (2011)	GSE26145	Affymetrix Human Exon 1.0 ST Array	6	6
	Osborne et al. (2008)	GSE10760	Affymetrix Human Genome U133A & B Arrays	19	30
	Arashiro et al. (2009)	GSE15090	Affymetrix Human Genome U133 Plus 2.0 Array	5	5
	Rahimov et al. (2012)	GSE36398	Affymetrix Human Gene 1.0 ST Array	13	12
	Tasca et al. (2012)	GSE26852	Illumina HumanHT- 12 V3.0 expression beadchip	8	7
RNA-Seq	Yao et al. (2014)	GSE56787**	Illumina HiSeq 2500	15	9
		GSE56787cellines		9	5
	Wang et al. (2018)	GSE115650	Illumina HiSeq 2500	33	9
	Banerji et al. (2018)	GSE123468_12***	Illumina HiSeq 2500	6	6
		GSE123468_16		6	6
van den Heuvel et al. (2022)	EGAD00001008337	Illumina HiSeq 4000 & Illumina NovaSeq 6000 & NextSeq 500	37	24	
Watt et al. (2021)	GSE138768 (DUX4)****	Illumina HiSeq 2500	4	4	
Σ				171	129

* More detailed information regarding patient and tissue characteristics, exclusion of samples and significant results can be found in *Supplementary Information S3*.

** GSE56787 included both cell lines and biopsies.

*** GSE123468 included two families (two sisters each as patient and control), which resulted in family-related batch effects in our statistical analysis due to strong genetic similarity in the families.

**** GSE138768 contains data from an artificial *DUX4* model and was therefore excluded from the overall calculation, but was contrasted for comparison.

- the text in figures is often too small. After the conversion of the figures into the main pdf, it's very difficult to read gene names, log-fold changes, or anything on the Fig3 schematic. We have made **Figure 3** clearer. A) Font size is larger; C) The app picture is larger; D) GO numbers have been removed and font size has been increased.

A. Type IV collagen genes and their possible influence on signal transduction

B. Random-effects model results for NMJ genes

C. Meta-FSHD app forest plot of COL4A3

D. Top 5 analyses results on CC pathways

GO term (↑)	P-value
collagen-containing extracellular matrix	2.531579 e-13
external encapsulating structure	1.528466 e-11
extracellular matrix	1.528466 e-11
cell-substrate junction	1.174668 e-10
focal adhesion	1.508024 e-10

GO term (↓)	P-value
T-tubule	1.911663 e-05
sarcolemma	3.392458 e-04
transmembrane transporter complex	6.176295 e-03
complex of collagen trimers	6.526375 e-03
transporter complex	1.258294 e-02

- Fig3 title is too strongly worded as a conclusion, when it's only a hypothesis based on (mild) gene expression changes.

We have changed it to: "Alterations at the NMJ and their possible influence on signal transduction." (Line number 469)

- it would be great to report gene numbers and fold enrichments for Fig3D Gene Ontology results.

For reasons of clarity, since we made the text in Figure 3 larger and there is a lot of information in the figure, we have made a note in the figure caption: "Further information (gene numbers and fold enrichments) can be found in Supplementary Table S3 for CC pathways." (Line numbers 486-487)

- Fig4C can you calculate a p-value for the overlaps?
We have included this information in the figure and we added in the figure caption: Adj. p-values (Holm's method) have been calculated using the R package SuperExactTest (Wang, M., Zhao, Y. & Zhang, B. Efficient Test and Visualization of Multi-Set Intersections. *Scientific Reports* 5, 16923 (2015).) (Line numbers 500-501)

- Circular plots in Fig5a and supps are not very clear. A 3-column format with gene name, logFC, annotations would be a lot easier to read.
In case of **Figure 5A**, we find this presentation very informative, as it clearly shows that the majority of genes are downregulated in terms of splicing in severely affected FSHD patients.
- Fig5B it would be helpful to report absolute numbers of AS events as well as their relative proportions.
We have included this information in the figure.

A. Expression of splicing associated genes

C. GO terms enriched in differentially spliced genes

D. Genes differentially spliced in FSHD

B. AS events in FSHD patients

E. ES and IR events in selected genes

I wasn't sure whether this analysis was done for only one of the selected RNA-seq datasets, could you clarify? If so why not include the others?

As FSHD is highly heterogeneous in terms of phenotype, muscle tissue and timing of pathogenic expression, we decided to perform a *sub-analysis* to refine and further examine the results of the meta-analysis. If the analysis was applied to all datasets, we would expect that the results would be less clear. GSE115650 was selected because it accounts for the heterogeneity of FSHD by categorizing pathogenic groups, which are additionally characterized by MRI data. Looking at all datasets, 1935 results are significant. Using this strategy of a *sub-analysis*, with a comparatively strong pathogenic expression (and neglecting mild forms), the number is 7400. The narrower focus provides a broader spectrum of results, which we have analysed with regard to the topics that emerged from the overall analysis.

- typos “sensitive analysis/analyses” for “sensitivity”; Fig1A has “für” instead of “for”; “enervation” for “innervation”.

We have changed this (line numbers 88, 570, 302).

Reviewer #2

Schatzl et al., present a meta-analysis of transcriptomes corresponding to FSHD patient muscle biopsies and cultured myoblasts and myotubes. They report mis-regulation of DUX4 target genes, metabolic, nuclear envelope, neuronal and spliceosome gene sets.

Unfortunately, the results and approach of this work are not novel, as numerous transcriptomic meta-analyses have been performed before for FSHD with similar conclusions. The manuscript also contains significant omissions and inaccuracies and in my opinion the methods are flawed.

Major omissions and inaccuracies:

- A PubMed search for 'FSHD transcriptomic meta-analysis' reveals 4 publications dating back 8 years:
 - Banerji et al., b-catenin is central to DUX4 driven network rewiring in facioscapulohumeral muscular dystrophy, *Journal of The Royal Society Interface*, 12(102):20140797, (2015).
 - Banerji et al., PAX7 target genes are globally repressed in FSHD skeletal muscle, *Nature Communications*, 8(2152) (2017).
 - Banerji et al., Skeletal muscle regeneration in Facioscapulohumeral muscular dystrophy is correlated with pathological severity, *Human Molecular Genetics*, 29(16):2746-2760, (2020).
 - Engquist et al., FSHD muscle shows perturbation in fibroadipogenic progenitor cells, mitochondrial function and alternative splicing independently of inflammation. *Hum. Mol. Genet.*, ddad175, (2024).

This last publication was published online in 2024. In this case we could not be aware of the information. Furthermore, the number of references is restricted and we made selections to the best of our knowledge.
- Despite the similarity in approach and significant overlap in overall conclusions, none of this highly relevant body of literature is discussed or even cited by the authors.

The following section has been inserted as an introduction to the discussion to take these missing sources into account:

"In the past, several studies have been conducted using mainly explorative meta-analytical approaches that yielded highly interesting results. For example, a link between *b-catenin* and *DUX4* was discovered (Banerji et al., 2015), and *PAX7* target genes were shown to be globally repressed in FSHD skeletal muscle (Banerji et al., 2017). Furthermore, skeletal muscle regeneration in FSHD was found to correlate with disease severity (Banerji et al., 2020). A recently published study also showed that FSHD muscle exhibits disruption of fibroadipogenic progenitor cells, mitochondrial function and alternative splicing independent of inflammation (Engquist et al., 2024). The latter underpins the results of our meta-analysis and shows the great importance of these aspects for the disease course in FSHD." (Line numbers 322-329)
- A key finding of prior FSHD meta-analyses has been suppression of PAX7 target genes in patient biopsies, this is not mentioned by the authors.

We have included this in the discussion part (line numbers 324-325).
- The authors also incorrectly claim novelty in the title and throughout the article.

The authors do not claim novelty of the topics but findings within the topics of research.

- The mis-regulation of nuclear envelope genes in FSHD has been known for many years and in fact was well demonstrated by transcriptomic meta-analysis in: Bakay et al., Nuclear envelope dystrophies show a transcriptional fingerprint suggesting disruption of Rb-MyoD pathways in muscle regeneration. *Brain* 2006 (doi: 10.1093/brain/awl023).

Schatzl et al., should be aware of this publication as it is associated with one of the data sets they considered (GSE3307), however, it is not cited or discussed in the manuscript.

The mis-regulation of the spliceosome is also well known in FSHD and has been shown in several publications over the last 8 years, for example: Rickard et al., Endogenous DUX4 expression in FSHD myotubes is sufficient to cause cell death and disrupts RNA splicing and cell migration pathways, *Hum. Mol. Genet.*, 2015 (<https://doi.org/10.1093/hmg/ddv315>).

There is a limit of permitted references in Nature. As already mentioned above, the authors do not claim novelty of the topics but findings within the topics of research.

There are three areas that we describe under "Confirmation of previous knowledge of FSHD pathology": *DUX4* activity (line numbers 104-117), mitochondrial deficits (line numbers 118-145), and relationship to the nuclear envelope (line numbers 146-164). The latter is supported by the following sources:

- Masny, P. S. et al. Localization of 4q35.2 to the nuclear periphery: Is FSHD a nuclear envelope disease? *Hum Mol Genet* 13, 1857–1871 (2004).

- Gaillard, M. C. et al. Analysis of the 4q35 chromatin organization reveals distinct long-range interactions in patients affected with Facio-Scapulo-Humeral Dystrophy. *Sci Rep* 9, (2019).

Other non-exhaustive specific inaccuracies include:

- 1. Line 164: ‘Interestingly, these neuronal aspects have not been described in FSHD, yet’ - This is not true mis-regulation of neuronal genes in FSHD has been known for many years, for example: Dandapt et al., Expression of the Human FSHD-Linked DUX4 Gene Induces Neurogenesis During Differentiation of Murine Embryonic Stem Cells. *Stem Cells Dev.*, 2013 (doi: 10.1089/scd.2012.0643).

This statement refers to: “(...) we became aware of genes within the CC category (in both up- and downregulated pathways) that are involved in pre- and postsynaptic processes, membranes and transitions, and neuronal processes of nerve projection.” The sentence refers to the previous sentence and should not be taken out of context. This is why the authors write “Interestingly, *these* neuronal aspects have not been described in FSHD, yet” rather than “Interestingly neuronal aspects have not been described in FSHD”.

- 2. Line 142: ‘skeletal muscle is almost exclusively affected in FSHD’ – This is not true, FSHD has prevalent extra-muscular manifestations, including sensorineural hearing loss (up to 30% in some studies) and retinal telangiectasia (up to 70% in some studies), this information is available in many reviews on FSHD.

This is why the authors write “almost”. Such statements (“retinal telangiectasia up to 70% in some studies (...)”, etc.) should not simply be taken out of context; This is an effect of *infantile* FSHD, which is estimated to occur in approximately 10% of disease carriers (Goselink et al., 2017). In fact, infantile FSHD is often correlated with extramuscular manifestations. In general, however, extramuscular symptoms are rather rare, with a few patients experiencing restrictive lung disease, cardiac conduction abnormalities, hearing loss, or retinal vasculopathy (Fitzsimons et al., 1987; Padberg et al., 1995; Laforet et al., 1998 ; Lutz et al., 2013; Scully et al., 2014; Lim et al., 2020c). However, to avoid any misunderstandings in this regard, we have

amended the sentence as follows: “This is highly interesting in terms of genome regulation, particularly with regard to the question of why skeletal muscles are so severely affected in FSHD.” (Line numbers 147-149)

Methodological Flaws:

The study selection criteria detailed in the supplementary document are contradictory in many regards and studies seem slightly cherry picked:

Methodologically, the meta-analysis was planned prospectively and registered in PROSPERO as a confirmatory analysis. In addition, the PRSIMA process was run through and the principles of conceptualization according to Ramasamy et al. (2008) were applied. For the studies’ inclusion, every single step was disclosed and described and quality assessment has been conducted. This is a very important basis for the methodological quality and validity of a meta-analysis. Furthermore, the meta-analysis was designed transparently in the spirit of open science (e.g. all analyses are also openly accessible in Github). We are not aware of any other analysis in FSHD research (including the meta-analytical approaches of Banerji et al. mentioned by Reviewer 2) that followed the same prospective and transparent approach in accordance with the respective guidelines (PRISMA2020, Ramasamy et al. (2008)) and the Cochrane Criteria for Systematic Reviews (Cochrane Handbook (vs 6.3, 2022)).

- On genotype: The authors state: ‘Because of the slightly different pathological background of the FSHD1 and FSHD2 disease types, datasets containing solely FSHD2 samples were removed. However, individual FSHD2 samples were retained since DUX4 expression has been described as a major feature of muscle wasting in both disease types⁶⁻⁸ and some FSHD2 samples of the included studies show strong DUX4 expression’ The distinction between FSHD1 and FSHD2 is never discussed anywhere in the main manuscript, which only refers to ‘FSHD’ and describes only the FSHD1 genotype in the introduction. This is an important omission given that the meta-analysis does include some (but notably not all) FSHD2 samples.

The authors have included FSHD2 in the introduction (line numbers 46-51).

The criteria for including an FSHD2 sample appears to be one of convenience, namely that the FSHD2 sample should be profiled alongside an FSHD1 sample and show high DUX4 expression. Given that a major finding of the authors is the up-regulation of DUX4 target genes, this selection criteria seems biased. One should either analyse all FSHD1 and FSHD2 samples or alternatively just select one genotype.

Moreover, in the case where both genotypes are considered there must be suitable adjustment of the differential expression (DE) analysis to account for genotype driven differences, this does not appear to have been performed.

In this regard, the original study design should be retained. The authors of earlier studies included both forms of the disease as FSHD1 and 2 show exactly the same phenotype. As there are few FSHD2 samples at the overall level of the analysis, these were therefore not excluded. This was the original view when designing the meta-analysis, which has subsequently been shown to be reasonable. Given the high rates of overlap between the datasets on which the vote-counting approach focuses, no differences are considered for FSHD1 and 2, only similarities, which therefore only underpins the FSHD characteristics.

- On muscle type: The authors state: ‘In contrast, the deltoid muscle is reported to be preserved or less affected^{21,22}. Therefore, studies that solely focused on the deltoid muscle were excluded from our meta-analysis.’ The muscle involvement of FSHD is highly heterogeneous, and several muscles are spared until late in the disease process, the muscles most often

associated with late involvement are deltoids and quadriceps (a summary with associated refs of muscle involvement in FSHD can be found in this review – Banerji and Zammit 2021 doi: 10.15252/emmm.202013695). To my knowledge there is no evidence that the deltoids are less involved than the quadriceps in FSHD. Hence if deltoids are excluded from the study on the basis of late involvement, we must also exclude quadriceps. Most muscle biopsies are quadriceps, so if this was done the meta-analysis would not really be possible. As with above the authors must analyse either all muscle groups, or restrict their analysis to one type (e.g., quads).

Again, if more than one muscle group is considered (as it currently is) the authors must adjust for muscle type specific gene expression in their differential expression analysis. Especially as this is well known to impact the expression of metabolic genes due to differences in slow/fast twitch fibre contributions, which are reported by the authors as altered in their FSHD meta-analysis.

In particular many of the data sets analysed contain biased and unequal muscle groups distributions between FSHD and control samples, which could drive spurious results in this analysis. Sample C6 in GSE56787, for example is the only TA sample in a data set otherwise comprising only quads and this has previously been shown to impact analysis (<https://doi.org/10.1038/s41467-017-01200-4>) – it is not clear if the authors considered this. The deltoid is often preserved (Statland, 2016, Pradhan, 2002). We do not claim the deltoid will not be affected by FSHD, but the involvement is simply less pronounced. Scapulothoracic arthrodesis, for instance, is the current surgical treatment for FSHD patients with severe winging and preserved deltoid muscle (Erren et al., 2022). In this regard, a study (GSE36398) that included biopsies of both the biceps and the deltoid muscle showed less involvement of the deltoid muscle in the published study "Transcriptional profiling in facioscapulohumeral muscular dystrophy to identify candidate biomarkers" (by Rahimov et al. 2012). In case of this highly selective muscle involvement, our decision was made by looking at the time course of disease progression. In this context, Lassche et al. (2021), demonstrated that reduced quadriceps specific force is an early feature of FSHD that occurs prior to the onset of fatty infiltration or clinical lower extremity weakness (DOI: 10.1002/mus.27074).

Furthermore, individual muscles were not excluded due to the very high variability of FSHD (tibialis anterior, deltoid, etc.). Thus, if the deltoid muscle was occasionally present in studies (eg. GSE26145), the data were not excluded to account for the large heterogeneity within FSHD.

- On mosaicism: Immortalised cell lines derived from mosaic patients were excluded from the meta-analysis with no justification, if these were biopsies, containing a mixture of pathological and healthy tissue I would accept this decision. However, these are clonal cell lines and are the closest one can get to a perfectly controlled system for FSHD, one line has an FSHD1 mutation the sister line does not but is otherwise isogenic, excluding these lines will only reduce inference.

We do not include mosaicism, as we looked at consistency with the other studies included. Moreover, immortalization is a process affecting several cellular characteristics (cell cycle, metabolism...) and thus adding an immense bias by itself. Immortalized cells are in no way a good system to identify general features of a disease.

- On myoblasts and myotubes: As with FSHD1 and FSHD2 genotypes, studies profiling myoblasts are included when it is convenient, namely when they are profiled alongside myotubes, but otherwise excluded, without clear justification:

'However, the analysis was extended to cell line datasets when both myoblast and corresponding myotube data were available, since two main stages of myogenesis from satellite cell to fibre are represented in these11-13. This finally allowed comparison of the respective data with those of the biopsy datasets.'

In order to observe the development of cells, myoblasts were included, which were differentiated into myotubes. Thus, one has two states of the same cell (which, in view of the high heterogeneity in FSHD, is characterized by individual characteristics).

- As an additional note, combining muscle biopsies and myoblast/myotube expression has many complications, which are not touched on in methods/discussion. These include immortalisation and the impact of this on telomeres/gene expression and differences in culture conditions (serum, differentiation protocol etc.).

It is well known that DUX4 is readily detected in MBs/MTs but not is almost never detected in biopsies and that these samples are very different in transcriptomic manifestation of the disease process and must be analysed separately (as has been done in previous meta-analyses).

Information from biopsies certainly provides the most information. However, in the meta-analysis, the data from myoblasts and their corresponding myotubes were used to refine/support the information from the biopsy datasets.

- The study GSE140261 is also omitted because it is a follow up study in a natural history of FSHD patients with no intervention, it seems odd to exclude additional data on these grounds, as they are essentially distinct technical replicates and may help minimise noise. Given that all of the cell line data sets contain technical replicates, which correspond to the same patient, and these are not excluded, why is exclusion done for the biopsy studies?

We have included independent datasets that are not linked in time, as otherwise additional dependencies should be incorporated into the analysis making it clearly more complex. The exclusion of the dataset was therefore a design decision.

Further comments on methods:

- The authors use limma for both microarray and RNA-seq DE analysis, this is suboptimal for RNA-seq where counts follow a negative binomial distribution a package such as DESeq2 should be used instead. Using limma will alter the distribution of significant DE genes in RNA-seq data and impact the results.

It is not true that DESeq2 is generally better than limma. It might be more straightforward to model counts by a negative binomial distribution, but by applying an appropriate transformation of the counts other approaches may be equally good in terms of their power/performance. Regarding limma this has been demonstrated in numerous studies, including:

> Ritchie et al. Limma powers differential expression analyses for RNA-sequencing and microarray studies. *Nucleic Acids Res* 43, e47 (2015).

> Tong, Y. The comparison of limma and DESeq2 in gene analysis. *E3S Web of Conferences* 271, 03058 (2021).

> Corchete et al. Systematic comparison and assessment of RNA-seq procedures for gene expression quantitative analysis. *Sci Rep* 10, (2020).

> Law et al. voom: precision weights unlock linear model analysis tools for RNA-seq read counts. *Genome Biology* vol. 15 <http://genomebiology.com/2014/15/2/R29> (2014).

- Random effects model:
Limma Voom was applied to the RNA-Seq datasets when the library sizes were very different between samples. Otherwise, if the ratio between the largest and smallest library size was no more than threefold, Limma Trend was used (according to Smyth et al., "limma: Linear Models for Microarray and RNA-Seq Data User's Guide," 2021). This results in a transformation leading to (at least an approximate) normal distribution, which is a necessary prerequisite for a meta-analysis that is based on means and SDs.
 - Vote counting approach:
As we used limma for differential expression analysis of microarray data we chose limma for RNA-Seq data preparation and analysis as well, since it required only minimal pipeline changes to switch between analyses for RNA-Seq and microarray experiments. Limma is transforming the read counts to the logarithmic scale and empirically estimating the relationship between mean and variance. The mean-variance trend is transformed into precision weights by the voom function, which is integrated into the analysis of log-transformed RNA-Seq counts using the same linear modelling commands as for microarrays. Thus, this approach provides a basis for comparability between the different technologies, since the same statistical tests are available for both types of data, with the same format of results and graphical representations.
2. Microarray and RNA-seq are very different technologies in terms of coverage and detection of the transcripts, this must be discussed in the paper as we are limited to genes represented in both.
The inclusion of a transcript was chosen so that this transcript must be present in at least 3 datasets. We have more than 3 microarray studies and more than 3 RNA-Seq studies. It does not have to be present in all of them. Accordingly, we aimed for good coverage.
 3. The authors discuss in the SI methods about GSE123468 being split into two data sets due to batch effects caused by different families. Batch effects are due to experimental set up, not genotype, these samples were processed in the same batch and so DE analysis should be done on the whole study adjusting for genotype as a covariate, not separately on the study cut in half, the latter approach is under-powered. Conversely, there are significant batch effects in one of the studies considered GSE36398, where disease status is confounded with experimental batch, the authors do not comment on this.
We have chosen the simpler, more pragmatic way. Overall, however, this has at most a very small influence on the power of the meta-analysis as a whole.
 4. Lines 212-216: The authors investigate the data set of Wang et al., 2019, but their findings seem confused: 'While the number of DEGs was already high when analyzing all samples together (1879), we found even more genes to be significantly misregulated when comparing the groups to the controls separately, with group 4 having the highest amount (8400). The reason for this could be the reduced variability within the groups, but also the increased disease pathology.' These 4 groups were defined by Wang et al., by how much their transcriptomes differed from control in the expression of a number of DUX4 targets, not pathological severity (that was a subsequent correlate), so it is not surprising that more DEGs are found in group 4 vs control compared to group 1 vs control, this is basically true by

construction. The authors then link this expected finding of more differentially expressed genes in group 4 back to pathological severity, in a circular argument.

The core of the statement was misunderstood here. In the paper, it is earlier described that the samples are characterized in terms of MRI and DUX4 expression. However, since the meta-analysis data show that FSHD is associated with changes in genes related to the nuclear envelope, the following sentence in this chapter is crucial: "Since there are so many DEGs in severely affected individuals, we wondered whether changes in genome organization cause these changes." The aim of this section is to take a closer look at the group of severely affected individuals with regard to the problems identified in the meta-analysis (which does not take into account the heterogeneity of FSHD).

In light of the above omissions, inaccuracies and methodological flaws, I cannot recommend this article for publication.

Reviewer #3

Schätzl et al. present a meta analysis of transcriptomic data prepared from patient-derived cell lines and / or tissue biopsies of individuals with predominantly facioscapulohumeral dystrophy (FSHD1). FSHD1 is a genetically intriguing disorder caused by an unusual non-coding, gene regulatory mechanism involving contraction of D4Z4 repeats on 4q35 and derepression of the DUX4 gene. The inverse of the D4Z4 repeat length and / or the DNA methylation status of the repeat usually, but do not always, correlate with disease severity. The aim of this study was to deeply investigate molecular pathways that are dysregulated in FSHD1 that may have been overlooked in previous studies. All data was extracted from public resources and spans 11 different studies carried out using technologies spanning 3' and exon arrays to RNA seq, between 2011 and 2022. The study is well conceived, well executed and a fantastic example of why FAIR principles as they apply to genomic data are so important. A weakness of the study is that due to the design, none of the novel findings, e.g. predicted alteration to neuromuscular junctions have been validated by orthogonal methods on patient tissue.

Major questions and comments:

- 1. Discussion on the involvement of dysregulated nuclear envelope proteins is confusing and requires more detail. Presumably the genes that are secondarily altered due to altered mechanotransduction are likely targets of the YAP / TAZ hippo pathway or other transcription factors which link with mechanosensitive pathways via the LINC complex. These targets may not show up when performing enrichment analyses using Biological Process, Molecular Function and Cellular Component ontologies. Further investigation of this by testing for enrichment of specific transcription factor involvement against TRANSFAC, JASPAR or similar databases would be of interest.

Regarding an analysis of transcription factor binding sites of genes that are potentially under control of the NETs TMEM38A and PLPP7 using TRANSFAC or JASPAR, we think it is an interesting additional analysis to be performed, but irrelevant to the point we are making: gene expression regulated through genome organization is one level above that of gene expression through transcription factors. 3D genome organization and chromatin packing allows or hinders transcription factor binding. For instance, there are topologically associated domains (TADs) that bring genes into close proximity that are under control of certain

transcription factors (reviewed in Campigli Di Giammartino et al. 2020), building hubs of transcription. Similarly, lamin associated domains (LADs) were traditionally associated with heterochromatin and gene silencing, but by now, it is commonly accepted that there is gene expression at the nuclear envelope. PLPP7 and TMEM38A pull specific genes towards the nuclear envelope and either promote or inhibit their expression (Robson et al. 2016, de las Heras et al. 2017), possibly through making them accessible/inaccessible to transcription factors. This doesn't rule out misregulation of transcription factors as a factor for altered gene expression, but it is a complex interaction making any clear conclusions at this point difficult.

- 2. The biological explanation for the lack of available spliceosomes in the discussion is not clear. In DM1, these factors are sequestered due to expression of the CUG repeat, what is the proposed mechanism at play in FSHD, do RNA foci form in severely affected individuals with FSHD? The relationship to FSHD and disruption to nonsense mediated mRNA decay (NMD) may be of interest. Feng Q, Snider L, Jagannathan S, Tawil R, van der Maarel SM, Tapscott SJ, et al. A feedback loop between nonsense-mediated decay and the retrogene DUX4 in facioscapulohumeral muscular dystrophy. Green R, editor. eLife. 2015 Jan 7;4:e04996. Shadle SC, Zhong JW, Campbell AE, Conerly ML, Jagannathan S, Wong CJ, et al. DUX4-induced dsRNA and MYC mRNA stabilization activate apoptotic pathways in human cell models of facioscapulohumeral dystrophy. PLOS Genetics. 2017 Mar 8;13(3):e1006658.

We have included the following information about splicing events: “Interestingly, nonsense-mediated-decay (NMD) has been shown previously to be downregulated in FSHD and intron retention leads mostly to mRNA-NMD (Feng et al. 2015; Shadle et al. 2017). This is an important regulatory level in gene expression governed through alternative splicing. In this context, the meta-analysis shows a high amount of significantly dysregulated RNA isoforms (*Supplementary Table S2*), which might be possible substrates for NMD. Interestingly, a key player of NMD, the exon junction complex component *EIF4A3*, which has been previously discussed by Shadle et al. (2017), is significantly upregulated in the meta-analysis results (STD log₂-FC (95% CI) = +0.651) and could be an important candidate for further research in this regard.” (Line numbers 291-299)

- 3. In the discussion the authors comment that spliceosomes assemble at every second or third splice site. This statement needs support of functional validation, e.g. measurement of spliceosome occupancy via RIP-seq or similar. With the available data it is not possible to conclude this vs. an altered splicing of cassette exons specifically driven by dysregulation of the spliceosome. Generalised lack of splicing machinery should lead to increased intron retention generally in the transcriptome, and potentially degradation of the majority of such transcripts by NMD (as noted above this may be compromised), however data as they are presented does not support this.

In addition to the statement about the possible role of NMD (see above), under the section “Components of the splicing machinery are downregulated in FSHD and result in the mis-splicing of muscle genes”, we have provided more detailed explanations as to why we hypothesize this in the discussion part:

“(…) In contrast, there was a general downregulation of alternative and constitutive splicing in group 4 (**Figure 5A**). This suggests a major disruption of the splicing machinery in strongly affected FSHD patients. We hypothesize that fewer spliceosomes assemble at splice sites due to the lower abundance of splicing factors, leading to many (constitutive and alternative) splice

sites being unused. This has to be experimentally validated in the future but is beyond the scope of this analysis.” (Line numbers 272-277)

Minor:

- 4. The manuscript reads in many places as a direct copy-paste from a thesis. The readability and clarity of the work could be improved by cutting down the volume of text and using a language style which is more direct. This is a suggestion only, I don't wish imply the author's chosen style is incorrect.
- 5. Page 7 lines 231 and 232: "...the majority is..." change to "... the majority were..."
We have changed this (line number 237).

Reviewers' comments:

Reviewer #1 (Remarks to the Author):

I'd like to thank the authors for engaging with my comments. However my major comment still stands.

The current filtering strategy based on log-fold changes certainly increases the power, but it also destroys control of the false discovery rate. Filtering that is not independent of the tested hypothesis should not be done. The paper that the authors cite in their response shows that FDR control is compromised. I cannot recommend publication until this fundamental methodological flaw is fixed. Choosing platform-specific expression thresholds is fine, the platforms measure very different things anyway so there's already platform bias, which the authors acknowledge in their response to reviewer 2, where they mention that they "aimed for good coverage" which sounds like an expression filter.

Reviewer #3 (Remarks to the Author):

In light of responses made to the comments I previously made on this manuscript I happy that they have been addressed. I have no further comments to make on the revised manuscript.

Reviewers' comments

Reviewer #1

I'd like to thank the authors for engaging with my comments. However my major comment still stands. The current filtering strategy based on log-fold changes certainly increases the power, but it also destroys control of the false discovery rate. Filtering that is not independent of the tested hypothesis should not be done. The paper that the authors cite in their response shows that FDR control is compromised. I cannot recommend publication until this fundamental methodological flaw is fixed. Choosing platform-specific expression thresholds is fine, the platforms measure very different things anyway so there's already platform bias, which the authors acknowledge in their response to reviewer 2, where they mention that they "aimed for good coverage" which sounds like an expression filter.

We agree that log fold change (FC) filters can change the distribution of the p-values under the null hypothesis and therefore a bias must be expected when calculating the false discovery rate (FDR) with the usual methods (van Iterson et al. 2010). We don't think that a bias, which can actually happen in many statistical analyses, especially if it is small "destroys" control of the FDR. To investigate the effect of FC filtering in our dataset, we undertook an analysis, that compares the results with and without filtering and put it in the Supplementary Material (*Supplementary Information S16*). The results of this analysis suggest that the true FDR for our 1935 genes is only slightly higher than 5% and might be in the range of 5-7%, which for us appears to be still a good control of the FDR.

We have added the following sentences to the paper: "(...) However, filtering can lead to a bias in the FDR calculation (van Iterson et al. 2010). The results in *Supplementary Information S16* suggest a true FDR of 5% to 7% for the selected 1935 genes. For comparison we have included the results of the unfiltered analysis in *Supplementary Table S2a and 2b*. (lines 578-581)".

Choosing platform specific expression thresholds might be fine, but from our point of view it seems difficult as we would have to find independent, cross-technology cut-offs to avoid an additional technical bias. This, in context of bias in general, in our point of view, is not necessarily less important as it also influences the overall results.

We agree that including only those genes that were measured in at least 3 studies could be interpreted as a kind of expression filter. We consider this to be unavoidable, as we believe that a meta-analysis with less than 3 studies would not be statistically meaningful. Requiring a prespecified number of studies is also included as an option in Step 23 of the guideline of Ramasamy et al. (2008).

We would also like to point out that a completely different approach, the "vote-counting approach" confirms the results of the random effects meta-analysis in terms of pathways and addressed topics (see *Supplementary Information S12* and *Supplementary Tables S4-S6*). In addition, we have investigated the robustness of our results by Leave-One-Out (LOO) analyses; see *Supplementary Information S6*. For us, this is a clear indication that the genes discussed in the paper are valid results. However, we understand the reviewer's point of view and hope that the additional analysis on the influence of the FC filter on the FDR will dispel the concerns expressed. If researchers look at the published material in the future and want to examine genes in more detail, the additions in the supplement will provide full information on filtered as well as unfiltered results.

Reviewer #3

In light of responses made to the comments I previously made on this manuscript I happy that they have been addressed. I have no further comments to make on the revised manuscript.

We would like to thank Reviewer 3 for the response.